# DiffPuter: Empowering Diffusion Models for Missing Data Imputation

**Hengrui Zhang**[1]     **Liancheng Fang**[1]     **Qitian Wu**[2]     **Philip S. Yu**[1]*

[1]Computer Science Department, University of Illinois at Chicago
[2]Eric and Wendy Schmidt Center, Broad Institute of MIT and Harvard
`{hzhan55,lfang87,psyu}@uic.edu`
`wuqitian@broadinstitute.org`

## ABSTRACT

Generative models play an important role in missing data imputation in that they aim to learn the joint distribution of full data. However, applying advanced deep generative models (such as Diffusion models) to missing data imputation is challenging due to 1) the inherent incompleteness of the training data and 2) the difficulty in performing conditional inference from unconditional generative models. To deal with these challenges, this paper introduces DiffPuter, a tailored diffusion model combined with the Expectation-Maximization (EM) algorithm for missing data imputation. DiffPuter iteratively trains a diffusion model to learn the joint distribution of missing and observed data and performs an accurate conditional sampling to update the missing values using a tailored reversed sampling strategy. Our theoretical analysis shows that DiffPuter's training step corresponds to the maximum likelihood estimation of data density (M-step), and its sampling step represents the Expected A Posteriori estimation of missing values (E-step). Extensive experiments across ten diverse datasets and comparisons with 17 different imputation methods demonstrate DiffPuter's superior performance. Notably, DiffPuter achieves an average improvement of 6.94% in MAE and 4.78% in RMSE compared to the most competitive existing method.

## 1 INTRODUCTION

In the field of data science and machine learning, missing data in tabular datasets is a common issue that can severely impair the performance of predictive models and the reliability of statistical analysis. Missing data can result from various factors, including data entry errors, non-responses in surveys, and system errors during data collection (Barnard & Meng, 1999; Lillard et al., 1986; Eckert et al., 2020). Properly handling missing data is essential, as improper treatment can lead to biased estimates, reduced statistical power, and invalid conclusions.

A plethora of work proposed over the past decades has propelled the development of missing data imputation research. Early classical methods often relied on partially observed statistical features to impute missing values or were based on conventional machine learning techniques, such as KNN (Pujianto et al., 2019), or simple parametric models, such as Bayesian models (Ma & Chen, 2018) or Gaussian Mixture Models (García-Laencina et al., 2010). With the advent of deep learning, recent research has primarily focused on predictive (Stekhoven & Bühlmann, 2012; Van Buuren & Karin, 2011; Kyono et al., 2021) or generative deep models (Yoon et al., 2018; Mattei & Frellsen, 2019; Richardson et al., 2020) for missing data imputation. Predictive models learn to predict the target entries conditioned on other observed entries, guided by masking mechanisms (Du et al., 2024) or graph regularization techniques (You et al., 2020; Zhong et al., 2023). By contrast, generative methods learn the joint distribution of missing entries and observed entries and aim to impute the missing data via conditional sampling (Richardson et al., 2020; Nazabal et al., 2020b; Mattei & Frellsen, 2019; Yoon et al., 2018; Ouyang et al., 2023; Zheng & Charoenphakdee, 2022). However, generative imputation methods still fall short compared to predictive methods even when employing state-of-the-art generative models (Kingma & Welling, 2013; Rezende & Mohamed,

---

*Corresponding author.

2015; Goodfellow et al., 2014; Ho et al., 2020). We think this is primarily due to the **incomplete likelihood** nature issue in missing data imputation: generative models need to estimate the joint distribution of missing data and observed data. However, since the missing data itself is unknown, there is an inherent error in the estimated data density. The classical Expectation-Maximization (EM) algorithm (Dempster et al., 1977) offers an elegant route to handle this issue, being capable of addressing the incomplete likelihood issue by iteratively refining the values of the missing data.

Integrating EM algorithms with generative models has been extensively studied (Richardson et al., 2020), however, its combination with Diffusion models (Ho et al., 2020), currently the most powerful generative models, is still unexplored. In the M-step, the diffusion models have been shown able to faithfully reconstruct the ground-truth distribution. However, in the E-step, it is usually considered challenging to utilize them to perform condition inference (e.g., predicting missing entries based on observed entries). This is because diffusion models directly model and generate the complete joint distribution of data across all dimensions simultaneously, lacking the flexibility found in VAE and GAN-based approaches (Ma et al., 2020; Peis et al., 2022b; Ma et al., 2018).

This paper introduces DIFFPUTER, a principled generative method for missing data imputation. DIFFPUTER explores a path that makes diffusion models compatible with the EM framework, allowing both E-step and M-step to be effective. Specifically: 1) In the M-step, DIFFPUTER employs a diffusion model to learn the joint distribution of the missing and observed data. With the powerful ability of diffusion models to learn tabular data distributions, in the M-step, DIFFPUTER learns a parameterized distribution that faithfully recovers the joint distribution of observed and missing entries. 2) In the E-step, DIFFPUTER uses the learned diffusion model to perform flexible and accurate conditional sampling by mixing the forward process for observed entries with the reverse process for missing entries. Theoretically, we show that DIFFPUTER's M-step corresponds to the maximum likelihood estimation of the data density, while its E-step represents the *Expected A Posteriori* (EAP) estimation of the missing values, conditioned on the observed values.

We conduct experiments[1] on nine benchmark tabular datasets containing both continuous and discrete features under various missing data scenarios. We compare the performance of DIFFPUTER with 17 competitive imputation methods from different categories. Experimental results demonstrate the superior performance of DIFFPUTER across all settings and on almost all datasets.

## 2 RELATED WORKS

**Iterative Methods for Missing Data Imputation.** Iterative imputation is a widely used approach due to its ability to continuously refine predictions of missing data, resulting in more accurate imputation outcomes. This iterative process is especially crucial for methods requiring an initial estimation of the missing data. The Expectation-Maximization (EM) algorithm (Dempster et al., 1977), a classical method, can be employed for missing data imputation. However, earlier applications often assume simple data distributions, such as mixtures of Gaussians for continuous data or Bernoulli and multinominal densities for discrete data (García-Laencina et al., 2010). These assumptions limit the imputation capabilities of these methods due to the restricted density estimation of simple distributions. The integration of EM with deep generative models remains underexplored. A closely related approach is MCFlow (Richardson et al., 2020), which iteratively imputes missing data using normalizing flows (Rezende & Mohamed, 2015). However, MCFlow focuses on recovering missing data through maximum likelihood rather than expectation, and its conditional imputation is achieved through soft regularization instead of precise sampling based on the conditional distribution. Beyond EM, the concept of iterative training is prevalent in state-of-the-art deep learning-based imputation methods. For instance, IGRM (Zhong et al., 2023) constructs a graph from all dataset samples and introduces the concept of friend networks, which are iteratively updated during the imputation learning process. HyperImpute (Jarrett et al., 2022) proposes an AutoML imputation method that iteratively refines both model selection and imputed values.

**Diffusion Models for Missing Data Imputation.** We are not the first to utilize diffusion models for missing data imputation. In computer vision, diffusion models have been widely applied in image inpainting, either in the data space (Lugmayr et al., 2022) or the latent space (Corneanu et al., 2024). In the tabular data area, TabCSDI (Zheng & Charoenphakdee, 2022) employs a conditional diffusion

---

[1]The code is available at https://github.com/hengruizhang98/DiffPuter.

model to learn the distribution of masked observed entries conditioned on the unmasked observed entries. MissDiff (Ouyang et al., 2023) uses a diffusion model to learn the density of tabular data with missing values by masking the observed entries. Although MissDiff was not originally intended for imputation, it can be easily adapted for this task. Other methods, despite claiming applicability for imputation, are trained on complete data and evaluated on incomplete testing data (Zhang et al., 2024; Jolicoeur-Martineau et al., 2024). This approach contradicts the focus of this study, where the training data itself contains missing values. Additionally, all the aforementioned methods use one-step imputations, which overlook the issue that missing data in the training set can lead to inaccurate data density estimation. By contrast, the proposed DIFFPUTER is the first to integrate a diffusion-based generative model into the EM framework. Additionally, we achieved accurate conditional sampling by mixing the forward and reverse processes of diffusion and demonstrated the effectiveness of this approach through theoretical analysis.

## 3 PRELIMINARIES

### 3.1 MISSING VALUE IMPUTATION FOR INCOMPLETE DATA

This paper addresses the missing value imputation task for incomplete data, where only partial data entries are observable during the training process. Formally, let the complete $d$-dimensional data be denoted as $\mathbf{x} \sim p_{\text{data}}(\mathbf{x}) \in \mathbb{R}^d$. For each data sample, $\mathbf{x}$, there is a binary mask $\mathbf{m} \in \{0, 1\}^d$ indicating the location of missing entries for $\mathbf{x}$. Let the subscript $k$ denote the $k$-th entry of a vector, then $m_k = 1$ stands for missing entries while $m_k = 0$ stands for observable entries.

We further use $\mathbf{x}^{\text{obs}}$ and $\mathbf{x}^{\text{mis}}$ to denote the observed data and missing data, respectively (i.e., $\mathbf{x} = (\mathbf{x}^{\text{obs}}, \mathbf{x}^{\text{mis}})$). Note that $\mathbf{x}^{\text{obs}}$ is the fixed ground-truth observation, while $\mathbf{x}^{\text{mis}}$ is conceptual and unknown, and we we aim to estimate it. The missing value imputation task aims to predict the missing entries $\mathbf{x}^{\text{mis}}$ based on the observed entries $\mathbf{x}^{\text{obs}}$.

**In-sample vs. Out-of-sample imputation.** The missing data imputation task can be categorized into two types: in-sample and out-of-sample. In-sample imputation means that the model only has to impute the missing entries in the training set, while out-of-sample imputation requires the model's capacity to generalize to the unseen data records without fitting its parameters again. Not all imputation methods can generalize to out-of-sample imputation tasks. For example, methods that directly treat the missing values as learnable parameters (Muzellec et al., 2020; Zhao et al., 2023) are hard to apply to unseen records. A desirable imputation method is expected to perform well on both in-sample and out-of-sample imputation tasks, and this paper studies both of the two settings.

### 3.2 FORMULATING MISSING DATA IMPUTATION WITH EXPECTATION-MAXIMIZATION

Treating $\mathbf{x}^{\text{obs}}$ as observed variables and $\mathbf{x}^{\text{mis}}$ as latent variables, with the estimated density of the complete data distribution parameterized as $p_{\boldsymbol{\theta}}(\mathbf{x})$, we can formulate the missing value imputation problem using the Expectation-Maximization (EM) algorithm. Specifically, when the complete data distribution $p_{\boldsymbol{\theta}}(\mathbf{x})$ is available, the optimal estimation of missing values is given by $\mathbf{x}^{\text{mis*}} = \mathbb{E}_{\mathbf{x}^{\text{mis}}} p(\mathbf{x}^{\text{mis}}|\mathbf{x}^{\text{obs}}, \boldsymbol{\theta})$. Conversely, when the missing entries $\mathbf{x}^{\text{mis}}$ are known, the density parameters can be optimized via maximum likelihood estimation: $\boldsymbol{\theta}^* = \arg\max_{\boldsymbol{\theta}} p(\mathbf{x}|\boldsymbol{\theta})$. Consequently, with the initial estimation of missing values $\mathbf{x}^{\text{mis}}$, the model parameters $\boldsymbol{\theta}$ and the missing value $\mathbf{x}^{\text{mis}}$ can be optimized by iteratively applying M-step and E-step:

- **M**aximization-step: Fix $\mathbf{x}^{\text{mis}}$, update $\boldsymbol{\theta}^* = \arg\max_{\boldsymbol{\theta}} p(\mathbf{x}|\boldsymbol{\theta}) = \arg\max_{\boldsymbol{\theta}} p_{\boldsymbol{\theta}}(\mathbf{x}^{\text{obs}}, \mathbf{x}^{\text{mis}})$.

- **E**xpectation-step: Fix $\boldsymbol{\theta}$, update $\mathbf{x}^{\text{mis*}} = \mathbb{E}_{\mathbf{x}^{\text{mis}}} p(\mathbf{x}^{\text{mis}}|\mathbf{x}^{\text{obs}}, \boldsymbol{\theta})$.

## 4 METHODOLOGY

In this section, we introduce DIFFPUTER- Iterative Missing Data Im**put**ation with **Diff**usion. Based on the Expectation-Maximization (EM) algorithm, DIFFPUTER updates the density parameter $\boldsymbol{\theta}$ and hidden variables $\mathbf{x}^{\text{mis}}$ in an iterative manner. Fig. 1 shows the overall architecture and training

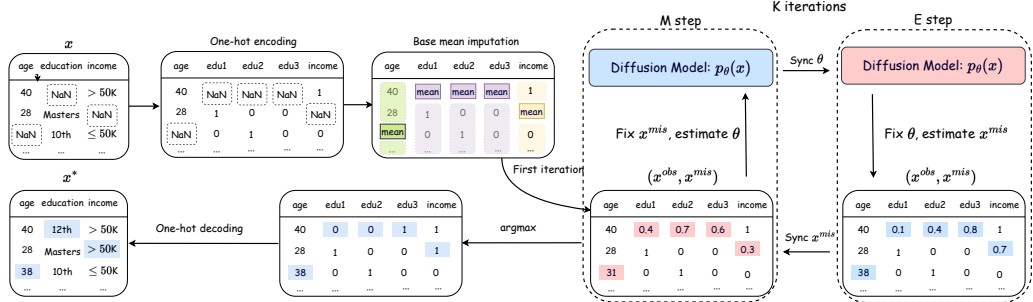

Figure 1: An overview of the architecture of the proposed DIFFPUTER. DIFFPUTER utilizes one-hot encoding to transform discrete variables into continuous ones and use the mean of observed values to initialize the missing entries. The EM algorithm alternates the process of 1) fixing $\mathbf{x}^{\mathrm{mis}}$ and estimate diffusion model parameter $\boldsymbol{\theta}$, 2) fixing $\boldsymbol{\theta}$ and estimate $\mathbf{x}^{\mathrm{mis}}$, for $K$ iterations. The final imputation result $\mathbf{x}^*$ is returned from the E-step of the last iteration.

process of DIFFPUTER: 1) The M-step fixes the missing entries $\mathbf{x}^{\mathrm{mis}}$, then a diffusion model is trained to estimate the density of the complete data distribution $p_{\boldsymbol{\theta}}(\mathbf{x}) = p_{\boldsymbol{\theta}}(\mathbf{x}^{\mathrm{obs}}, \mathbf{x}^{\mathrm{mis}})$; 2) The E-step fixes the model parameters $\boldsymbol{\theta}$, then we update the missing entries $\mathbf{x}^{\mathrm{mis}}$ via the reverse process of the learned diffusion model $p_{\boldsymbol{\theta}}(\mathbf{x})$. The above two steps are executed iteratively until convergence. The following sections introduce the M-step and E-step of DIFFPUTER, respectively. To avoid confusion, we use $\mathbf{x}, \mathbf{x}_t, \mathbf{x}^{\mathrm{obs}}$, etc. to denote samples from real data, $\tilde{\mathbf{x}}, \tilde{\mathbf{x}}_t, \tilde{\mathbf{x}}^{\mathrm{obs}}$, etc. to denote samples obtained by the model $\boldsymbol{\theta}$, while $\hat{\mathbf{x}}, \hat{\mathbf{x}}_t, \hat{\mathbf{x}}^{\mathrm{obs}}$, etc. to denote the specific values of variables.

## 4.1 M-STEP: DENSITY ESTIMATION WITH DIFFUSION MODELS

Given an estimation of complete data $\mathbf{x} = (\mathbf{x}^{\mathrm{obs}}, \mathbf{x}^{\mathrm{mis}})$, M-step aims to learn the density of $\mathbf{x}$, parameterized by model $\boldsymbol{\theta}$, i.e., $p_{\boldsymbol{\theta}}(\mathbf{x})$. Inspired by the impressive generative modeling capacity of diffusion models (Song et al., 2021b; Karras et al., 2022), DIFFPUTER learns $p_{\boldsymbol{\theta}}(\mathbf{x})$ through a diffusion process, which consists of a forward process that gradually adds Gaussian noises of increasing scales to $\mathbf{x}$, and a reverse process that recovers the clean data from the noisy one:

$$\mathbf{x}_t = \mathbf{x}_0 + \sigma(t)\boldsymbol{\varepsilon}, \ \boldsymbol{\varepsilon} \sim \mathcal{N}(\mathbf{0}, \mathbf{I}), \qquad \text{(Forward Process)} \qquad (1)$$

$$\mathrm{d}\mathbf{x}_t = -2\dot{\sigma}(t)\sigma(t)\nabla_{\mathbf{x}_t} \log p(\mathbf{x}_t)\mathrm{d}t + \sqrt{2\dot{\sigma}(t)\sigma(t)}\mathrm{d}\boldsymbol{\omega}_t, \qquad \text{(Reverse Process)} \qquad (2)$$

In the forward process, $\mathbf{x}_0 = \mathbf{x} = (\mathbf{x}^{\mathrm{obs}}, \mathbf{x}^{\mathrm{mis}})$ is the currently estimated data at time 0, and $\mathbf{x}_t$ is the diffused data at time $t$. $\sigma(t) = t$ is the noise level (and $\dot{\sigma}(t)$ is its derivative w.r.t. $t$), i.e., the standard deviation of Gaussian noise, at time $t$. The forward process has defined a series of data distribution $p(\mathbf{x}_t) = \int_{\mathbf{x}_0} p(\mathbf{x}_t|\mathbf{x}_0)p(\mathbf{x}_0)\mathrm{d}\mathbf{x}_0$, and $p(\mathbf{x}_0) = p(\mathbf{x})$. Note that when restricting the mean of $\mathbf{x}_0$ to 0 and keeping the variance of $\mathbf{x}_0$ small (e.g., via standardization), $p(\mathbf{x}_t)$ approaches a tractable prior distribution $\pi(\mathbf{x})$ at $t = T$ when $\sigma(T)$ is large enough, meaning $p(\mathbf{x}_T) \approx \pi(\mathbf{x})$ (Song et al., 2021a). In our formulation in Eq. 1, $p(\mathbf{x}_T) \approx \pi(\mathbf{x}) = \mathcal{N}(\mathbf{0}, \sigma^2(T)\mathbf{I})$.

In the reverse process, $\nabla_{\mathbf{x}_t} \log p(\mathbf{x}_t)$ is the gradient of $\mathbf{x}_t$'s log-probability w.r.t., to $\mathbf{x}_t$, and is also known as the *score function*. $\boldsymbol{\omega}_t$ is the standard Wiener process. The model is trained by (conditional) score matching (Song et al., 2021b), which utilizes a neural network $\boldsymbol{\epsilon}_{\boldsymbol{\theta}}(\mathbf{x}_t, t)$ (called denoising/score network) to approximate the conditional score-function $\nabla_{\mathbf{x}_t} \log p(\mathbf{x}_t|\mathbf{x}_0)$, which in expectation approximates $\nabla_{\mathbf{x}_t} \log p(\mathbf{x}_t)$:

$$\mathcal{L}_{\mathrm{SM}} = \mathbb{E}_{\mathbf{x}_0 \sim p(\mathbf{x}_0)} \mathbb{E}_{t \sim p(t)} \mathbb{E}_{\boldsymbol{\varepsilon} \sim \mathcal{N}(\mathbf{0}, \mathbf{I})} \|\boldsymbol{\epsilon}_{\boldsymbol{\theta}}(\mathbf{x}_t, t) - \nabla_{\mathbf{x}_t} \log p(\mathbf{x}_t|\mathbf{x}_0)\|_2^2, \ \text{where } \mathbf{x}_t = \mathbf{x}_0 + \sigma(t)\boldsymbol{\varepsilon}. \ (3)$$

Since the score of conditional distribution has analytical solutions, i.e., $\nabla_{\mathbf{x}_t} \log p(\mathbf{x}_t|\mathbf{x}_0) = \frac{\nabla_{\mathbf{x}_t} p(\mathbf{x}_t|\mathbf{x}_0)}{p(\mathbf{x}_t|\mathbf{x}_0)} = -\frac{\mathbf{x}_t - \mathbf{x}_0}{\sigma^2(t)} = -\frac{\boldsymbol{\varepsilon}}{\sigma(t)}$, Eq. 3 can be interpreted as training a neural network $\boldsymbol{\epsilon}_{\boldsymbol{\theta}}(\mathbf{x}_t, t)$ to

approximate the scaled noise. Therefore, $\epsilon_{\boldsymbol{\theta}}(\mathbf{x}_t, t)$ is also known as the denoising function, and in this paper, it is implemented as a five-layer MLP (see Appendix D.4).

**Remark 1.** *Starting from the prior distribution $\mathbf{x}_T \sim \pi(\mathbf{x})$, and apply the reverse process in Eq. 2 with $\nabla_{\mathbf{x}_t} \log p(\mathbf{x}_t)$ replaced with $\epsilon_{\boldsymbol{\theta}}(\mathbf{x}_t, t)$, we obtain a series of distributions $p_{\boldsymbol{\theta}}(\mathbf{x}_t), \forall t \in [0, T]$*

**Remark 2.** (Corollary 1 in (Song et al., 2021a)) *Let $p(\mathbf{x}) = p(\mathbf{x}_0)$ be the data distribution, and $p_{\boldsymbol{\theta}}(\mathbf{x}) = p_{\boldsymbol{\theta}}(\mathbf{x}_0)$ be the marginal data distribution obtained from the reverse process, then the score-matching loss function in Eq. 3 is an upper bound of the negative-log-likelihood of real data $\boldsymbol{x} \sim p(\boldsymbol{x})$ over $\boldsymbol{\theta}$. Formally,*

$$-\mathbb{E}_{p(\mathbf{x})}[\log p_{\boldsymbol{\theta}}(\mathbf{x})] \leq \mathcal{L}_{\text{SM}}(\boldsymbol{\theta}) + Const, \tag{4}$$

*where $Const$ is a constant independent of $\boldsymbol{\theta}$.*

Remark 2 indicates that the $\boldsymbol{\theta}$ optimized by Eq. 3 approximates the maximum likelihood estimation of data distribution $p(\mathbf{x})$. Consequently, $p_{\boldsymbol{\theta}}(\boldsymbol{x})$ approximates $p(\mathbf{x})$ accurately with enough capacity. The algorithmic illustration of DIFFPUTER's M-step is summarized in Algorithm 1.

## 4.2 E-STEP: MISSING DATA IMPUTATION WITH A LEARNED DIFFUSION MODEL

Given the current estimation of data distribution $p_{\boldsymbol{\theta}}(\mathbf{x})$, the E-step aims to obtain the distribution of complete data, conditional on the observed values, i.e., $p_{\boldsymbol{\theta}}(\mathbf{x}|\mathbf{x}^{\text{obs}})$, such that the estimated complete data $\mathbf{x}^*$ can be updated by taking the expectation, i.e., $\mathbf{x}^* = \mathbb{E}_{\mathbf{x}} p_{\boldsymbol{\theta}}(\mathbf{x}|\mathbf{x}^{\text{obs}})$.

When there is an explicit density function for $p_{\boldsymbol{\theta}}(\mathbf{x})$, or when the conditional distribution $p_{\boldsymbol{\theta}}(\mathbf{x}|\mathbf{x}^{\text{obs}})$ is tractable (e.g., can be sampled), computing $\mathbb{E}_{\mathbf{x}} p(\mathbf{x}|\mathbf{x}^{\text{obs}}, \boldsymbol{\theta})$ becomes feasible. While most deep generative models such as VAEs and GANs support convenient unconditional sampling from $p_{\boldsymbol{\theta}}(\mathbf{x})$, they do not naturally support conditional sampling, e.g., $\tilde{\mathbf{x}} \sim p_{\boldsymbol{\theta}}(\mathbf{x}|\mathbf{x}^{\text{obs}})$. Luckily, since the diffusion model preserves the size and location of features in both the forward diffusion process and the reverse denoising process, it offers a convenient and accurate solution to perform conditional sampling $p_{\boldsymbol{\theta}}(\mathbf{x}|\mathbf{x}^{\text{obs}})$ from an unconditional model $p_{\boldsymbol{\theta}}(\mathbf{x}) = p_{\boldsymbol{\theta}}(\mathbf{x}^{\text{obs}}, \mathbf{x}^{\text{mis}})$.

Specifically, let $\mathbf{x}$ be the data to impute, $\mathbf{x}^{\text{obs}} = \hat{\mathbf{x}}_0^{\text{obs}}$ be the values of observed entries, $\mathbf{m}$ be the $0/1$ indicators of the location of missing entries, and $\tilde{\mathbf{x}}_t$ be the imputed data at time $t$. Then, we can obtain the imputed data at time $t - \Delta t$ via combining the observed entries from the forward process on $\mathbf{x}_0 = \mathbf{x}$, and the missing entries from the reverse process on $\tilde{\mathbf{x}}_t$ from the prior step $t$ (Lugmayr et al., 2022; Song et al., 2021b):

$$\mathbf{x}_{t-\Delta t}^{\text{foward}} = \mathbf{x} + \sigma(t - \Delta t) \cdot \boldsymbol{\varepsilon}, \text{ where } \boldsymbol{\varepsilon} \sim \mathcal{N}(\mathbf{0}, \mathbf{I}), \quad \text{(Forward for observed entries)} \quad (5)$$

$$\mathbf{x}_{t-\Delta t}^{\text{reverse}} = \tilde{\mathbf{x}}_t + \int_t^{t-\Delta t} \mathrm{d}\tilde{\mathbf{x}}_t, \text{ where } \mathrm{d}\tilde{\mathbf{x}}_t \text{ is defined in Eq. 2} \quad \text{(Reverse for missing entries)} \quad (6)$$

$$\tilde{\mathbf{x}}_{t-\Delta t} = (1 - \mathbf{m}) \odot \mathbf{x}_{t-\Delta t}^{\text{forward}} + \mathbf{m} \odot \mathbf{x}_{t-\Delta t}^{\text{reverse}}. \tag{7}$$

Based on the above process, starting at a random noise from the maximum time $T$, i.e., $\tilde{\mathbf{x}}_T \sim \mathcal{N}(\mathbf{0}, \sigma^2(T)\mathbf{I})$, we can obtain a reconstructed $\tilde{\mathbf{x}}_0$, such that the observed entries of $\tilde{\mathbf{x}}_0$ are the same as those of $\mathbf{x}_0$, i.e., $\tilde{\mathbf{x}}_0^{\text{obs}} = \hat{\mathbf{x}}_0^{\text{obs}}$. In Theorem 1, we prove that via the algorithm above, the obtained $\tilde{\mathbf{x}}_0$ is sampled from the desired conditional distribution, i.e., $\tilde{\mathbf{x}}_0 \sim p_{\boldsymbol{\theta}}(\mathbf{x}|\mathbf{x}^{\text{obs}} = \hat{\mathbf{x}}_0^{\text{obs}})$.

**Theorem 1.** *Let $\tilde{\mathbf{x}}_T$ be a sample from the prior distribution $\pi(\mathbf{x}) = \mathcal{N}(\mathbf{0}, \sigma^2(T)\mathbf{I})$, $\mathbf{x}$ be the data to impute, and the known entries of $\mathbf{x}$ are denoted by $\mathbf{x}^{\text{obs}} = \hat{\mathbf{x}}_0^{\text{obs}}$. The score function $\nabla_{\mathbf{x}_t} \log p(\mathbf{x}_t)$ is approximated by neural network $\epsilon_{\boldsymbol{\theta}}(\mathbf{x}_t, t)$ . Applying Eq. 5, Eq. 6, and Eq. 7 iteratively from $t = T \gg 0$ until $t = 0$ with $\Delta t \to 0$, then $\tilde{\mathbf{x}}_0$ is a sample from $p_{\boldsymbol{\theta}}(\mathbf{x})$, under the condition that its observed entries $\tilde{\mathbf{x}}_0^{\text{obs}} = \hat{\mathbf{x}}_0^{\text{obs}}$. Formally,*

$$\tilde{\mathbf{x}}_0 \sim p_{\boldsymbol{\theta}}(\mathbf{x}|\mathbf{x}^{\text{obs}} = \hat{\mathbf{x}}_0^{\text{obs}}) \tag{8}$$

See proof in Appendix B.1. Theorem 1 demonstrates that with a learned diffusion model $\boldsymbol{\theta}$, we are able to obtain samples exactly from the conditional distribution $p_{\boldsymbol{\theta}}(\mathbf{x}|\mathbf{x}^{\text{obs}})$ through the aforementioned imputation process. For inference, we use Monte Carlo estimation to compute the expectation of the missing values, i.e., $\mathbf{x}^* = \mathbb{E}_{\mathbf{x}} p_{\boldsymbol{\theta}}(\mathbf{x}|\mathbf{x}^{\text{obs}}) \approx \sum_{j=1}^N \tilde{\mathbf{x}}_0^{(j)}/N$.

**Discretization.** The above imputing process involves recovering $\tilde{\mathbf{x}}_t$ continuously from $t = T$ to $t = 0$, which is infeasible in practice. In implementation, we discretize the process via $M + 1$ discrete descending timesteps $t_M, t_{M-1}, \cdots, t_i, \cdots, t_0$, where $t_M = T, t_0 = 0$. Therefore, starting from $\tilde{\mathbf{x}}_{t_M} \sim p_{\boldsymbol{\theta}}(\mathbf{x}_{t_M} | \mathbf{x}^{\text{obs}} = \hat{\mathbf{x}}_0^{\text{obs}})$, we obtain $p_{\boldsymbol{\theta}}(\mathbf{x}_{t_i} | \mathbf{x}^{\text{obs}} = \hat{\mathbf{x}}_0^{\text{obs}})$ from $i = M - 1$ to $i = 0$.

---

**Algorithm 1:** M-step: Density Estimation using Diffusion Model

---

**Input:** Data samples from $p(\mathbf{x})$.
**Output:** Score network $\boldsymbol{\epsilon}_{\boldsymbol{\theta}}$

1 **while** *not converging* **do**
2      Sample $\mathbf{x} \sim p(\mathbf{x}) \in \mathbb{R}^d$
3      Sample $t \sim p(t)$
4      Sample $\boldsymbol{\varepsilon} \sim \mathcal{N}(\mathbf{0}, \mathbf{I}) \in \mathbb{R}^d$
5      $\mathbf{x}_0 = \mathbf{x}$
6      $\mathbf{x}_t = \mathbf{x}_0 + \sigma(t) \cdot \boldsymbol{\varepsilon}$
7      $\ell(\boldsymbol{\theta}) = \|\boldsymbol{\epsilon}_{\boldsymbol{\theta}}(\mathbf{x}_t, t) - \frac{-\boldsymbol{\varepsilon}}{\sigma(t)}\|_2^2$
8      Update $\boldsymbol{\theta}$ via Adam optimizer

---

**Algorithm 2:** E-step: Missing Data Imputation

---

**Input:** Score network $\boldsymbol{\epsilon}_{\boldsymbol{\theta}}(\mathbf{x}_t, t)$. Data with missing values $\mathbf{x} \in \mathbb{R}^d$, mask $\mathbf{m} \in \{0, 1\}^d$. Number of samples $N$. Number of sampling steps $M$.
**Output:** Imputed data $\mathbf{x}^*$

1 **for** $j \leftarrow 1$ **to** $N$ **do**
2      Sample $\tilde{\mathbf{x}}_{t_M}^{(j)} \sim \mathcal{N}(\mathbf{0}, \sigma^2(t_M)\mathbf{I})$
3      **for** $i \leftarrow M$ **to** $1$ **do**
4          $\mathbf{x}_{t_{i-1}}^{\text{forward},(j)} = \mathbf{x} + \sigma(t_{i-1}) \cdot \boldsymbol{\varepsilon}$
5          $\mathbf{x}_{t_{i-1}}^{\text{reverse},(j)} = \tilde{\mathbf{x}}_{t_i}^{(j)} + \int_{t_i}^{t_{i-1}} \mathrm{d}\tilde{\mathbf{x}}_{t_i}^{(j)}$
6          $\tilde{\mathbf{x}}_{t_{i-1}}^{(j)} = \mathbf{m} \odot \mathbf{x}_{t_{i-1}}^{\text{forward},(j)} + (1-\mathbf{m}) \odot \mathbf{x}_{t_{i-1}}^{\text{reverse},(j)}$
7 $\mathbf{x}^* = \sum_{j=1}^{N} \tilde{\mathbf{x}}_{t_0}^{(j)}/N = \sum_{j=1}^{N} \tilde{\mathbf{x}}_0^{(j)}/N$

---

Since the desired imputed data $\mathbf{x}^{\text{mis}*}$ is the expectation, i.e., $\mathbf{x}^{\text{mis}*} = \mathbb{E}_{\mathbf{x}^{\text{mis}}} p(\mathbf{x}^{\text{mis}} | \mathbf{x}^{\text{obs}} = \hat{\mathbf{x}}_0^{\text{obs}})$, we sample $\tilde{\mathbf{x}}_0^{\text{mis}}$ for $N$ times, and take the average value as the imputed $\mathbf{x}^{\text{mis}*}$. The algorithmic illustration of DIFFPUTER's E-step is summarized in Algorithm 2.

### 4.3 IMPLEMENTATIONS

**Data Processing.** Diffusion models are inherently suited for continuous data but not for discrete data. Given that real-world tabular data often contains both types of data, we use one-hot encoding to convert each dimension of discrete data into multi-dimensional 0/1 encoding, thereby treating it as continuous data . After that, we perform standardization on each column of data, ensuring that it has a mean of 0 and a variance of 1. The final imputation performance is measured by the difference between the predicted value and the ground-truth missing value **after standardization**.

**Initialization of Missing Data.** The execution of the EM algorithm requires the initialized values of missing entries $\mathbf{x}^{\text{mis}(0)}$, which might have a huge impact on the model's convergence. For simplicity, we initialize the missing entries of each column to be the mean of the column's observed values, equivalent to setting $\mathbf{x}^{\text{mis}(0)} = 0$ everywhere (since the data has been standardized).

**Training.** To obtain a more accurate estimation of complete data $\mathbf{x}$, DIFFPUTER iteratively executes the M-step and E-step. To be specific, let $\mathbf{x}^{(k)} = (\mathbf{x}^{\text{obs}(k)}, \mathbf{x}^{\text{obs}(k)}) = (\mathbf{x}^{\text{obs}}, \mathbf{x}^{\text{obs}(k)})$ be the estimation of complete data at $k$-th iteration, $\boldsymbol{\theta}^{(k)}$ be the diffusion model's parameters at $k$-th iteration, we write $\boldsymbol{\theta}^{(k+1)}$ as a function of $\mathbf{x}^{(k)}$, i.e., $\boldsymbol{\theta}^{(k+1)} = $ M-step $(\mathbf{x}^{(k)})$, and $\mathbf{x}^{(k+1)}$ as a function of $\boldsymbol{\theta}^{(k+1)}$ and $\mathbf{x}^{(k)}$, i.e., $\mathbf{x}^{(k+1)} = $ E-step $(\mathbf{x}^{(k)}, \boldsymbol{\theta}^{(k+1)})$. Therefore, with the initialized $\mathbf{x}^{(0)}$ and the maximum iteration $K$, we are able to obtain $\mathbf{x}^{(k)}$ from $k = 1$ to $K$.

**Inference.** For in-sample imputation, the imputed values are obtained iteratively during the training process. For out-of-sample imputation, with the observed incomplete data $\mathbf{x}$, the mask $\mathbf{m}$, and the trained score network $\boldsymbol{\epsilon}_{\boldsymbol{\theta}}$, we directly apply the E-step once in Algorithm 2 to obtain the imputation values $\mathbf{x}^*$.

## 5 EXPERIMENTS

In this section, we conduct experiments to study the efficacy of the proposed DIFFPUTER in missing data imputation tasks.

## 5.1 EXPERIMENTAL SETTINGS

**Datasets.** We evaluate the proposed DIFFPUTER on ten public real-world datasets of varying scales. We consider five datasets of only continuous features: California, Letter, Gesture, Magic, and Bean, and four datasets of both continuous and discrete features: Adult, Default, Shoppers, and News. The detailed information of these datasets is presented in Appendix D.2. Following previous works (Muzellec et al., 2020; Zhao et al., 2023), we study three missing mechanisms: MCAR, MAR, and MNAR. Differences between the three settings are in Appendix D.3. In this section, we only report the performance in the MCAR setting, while the results of the other two settings are in Appendix E. In the main experiments, we set the missing rate as $r = 30\%$. For each dataset, we generate 10 masks according to the missing mechanism and report the mean and standard deviation of the imputing performance.

**Baselines.** We compare DIFFPUTER with 16 powerful imputation methods from different categories: 1) Distribution-matching methods based on optimal transport, including TDM (Zhao et al., 2023) and MOT (Muzellec et al., 2020). 2) Graph-based imputation methods, including GRAPE (You et al., 2020): a pure bipartite graph-based framework for data imputation, and IGRM (Zhong et al., 2023): a graph-based imputation method that iteratively reconstructs the friendship network. 3) Iterative methods, including EM with multivariate Gaussian priors (García-Laencina et al., 2010), MICE (Van Buuren & Karin, 2011), MIRACLE (Kyono et al., 2021), SoftImpute (Hastie et al., 2015), and MissForest (Stekhoven & Bühlmann, 2012). 4) Deep generative models, including MIWAE (Mattei & Frellsen, 2019), GAIN (Yoon et al., 2018), MCFlow (Richardson et al., 2020), MissDiff (Ouyang et al., 2023) and TabCSDI (Zheng & Charoenphakdee, 2022). It is worth noting that MissDiff and TabCSDI are also based on diffusion. We also compare with two recent SOTA imputing methods ReMasker (Du et al., 2024) and HyperImpute (Jarrett et al., 2022).

**Evaluation Protocols.** For each dataset, we use $70\%$ as the training set, and the remaining $30\%$ as the testing set. All methods are trained on the training set. The imputation is applied to both the missing values in the training set and the testing set. Consequently, the imputation of the training set is the 'in-sample' setting, while imputing the testing set is the 'out-of-sample' setting. The performance of DIFFPUTER is evaluated by the divergence between the predicted values and ground-truth values of missing entries. For continuous features, we use Mean Absolute Error (MAE) and Root Mean Squared Error (RMSE), while for discrete features, we use Accuracy. Note that both the MAE and RMSE are calculated based on the input data after standardization (zero mean and unit variance). The implementation details and hyperparameter settings are presented in Appendix D.4.

## 5.2 MAIN RESULTS (MASK COMPLETELY AT RANDOM)

We first evaluate DIFFPUTER's performance in the in-sample imputation task. Figure 2 compares the performance of different imputation methods regarding continuous columns using MAE and RMSE. Table 1 compares the performance of different methods regarding discrete columns using accuracy. Our observations on these experimental results are summarized as follows.

**Superior performance of DIFFPUTER.** Across all datasets, DIFFPUTER provides high-quality imputation results, matching the best methods on some datasets and significantly outperforming the second-best methods on the remaining datasets. Compared with the other two diffusion-based methods, MissDiff and TabCSDI, the substantial performance advantage of DIFFPUTER convincingly demonstrates the correctness and superiority of combining the EM algorithm with Diffusion.

**Traditional machine learning methods are still powerful imputors.** A well-recognized claim in tabular data machine learning is that traditional machine learning methods can sometimes be better than deep learning methods (Lalande & Doya, 2022; Suh & Song, 2023; Jolicoeur-Martineau et al., 2024), and we have similar observations in missing data imputation. For example, the simple EM algorithm under the mixture-Gaussian assumption and the KNN algorithm give decent imputation performance and outperform a lot of early deep generative imputation methods.

**The preference of different types of algorithms for heterogeneous tabular data.** Another interesting finding is that discriminative methods (e.g., Remasker, GRAPE, MOT) and generative methods (e.g., DIFFPUTER) have different preferences for continuous data and discrete data. Overall, generative models appear to be more effective at imputing continuous columns. As shown in Figure 2, our model significantly outperforms these discriminative models. Conversely, discriminative models

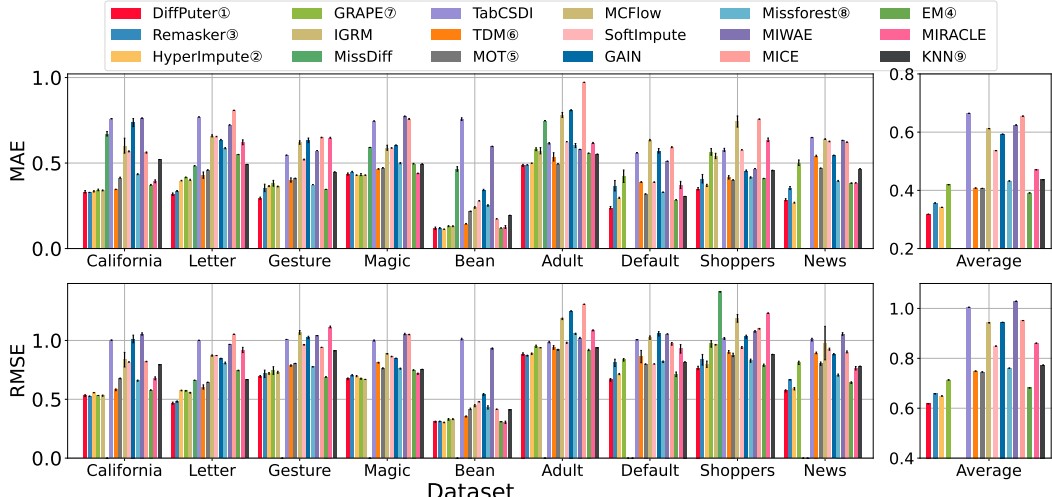

Figure 2: MCAR, In-sample imputation performance on **continuous columns**: Comparing DIFF-PUTER with **17 baselines** on imputing continuous data on all the **nine datasets**. A blank column indicates that the method fails or gets out-of-memory for that dataset. DIFFPUTER outperforms the most competitive baseline method by $6.94\%$ (MAE score) and $4.78\%$ (RMSE score) by average. The circled number after the model name denotes its ranking among all methods.

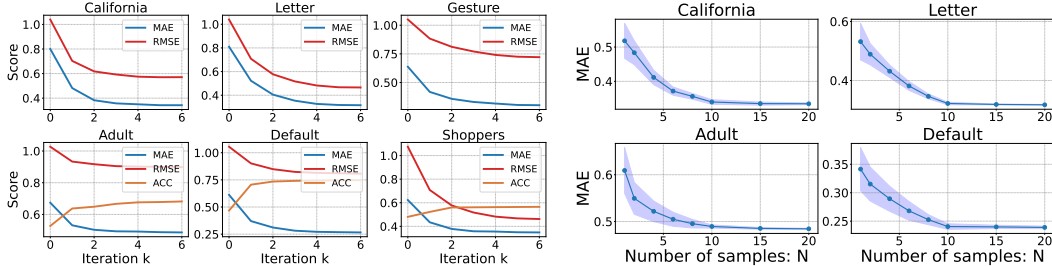

Figure 3: Impacts of the number of EM iterations. DIFFPUTER's performance steadily improves as the number of EM interactions increases.

Figure 4: Impacts of the number of sampled imputations per iteration. A very small $N$ leads to poor performance and large variance.

are more proficient at imputing discrete data: in Table 1, the best discriminative model, Remasker, can achieve results comparable to our method on almost all datasets.

Next, we compare the performance in the out-of-sample imputation tasks. Noting that some imputation methods are specifically designed for in-sample imputation and cannot be applied to the out-of-sample setting, the number of baselines in this task is significantly reduced. Table 6 in Appendix E.1 compares the MAEs and RMSEs in the OOS imputation task. Comparing it with the results of in-sample imputation, we can easily observe that some methods exhibit significant performance differences between the two settings. For example, graph-based methods GRAPE and IGRM perform well in in-sample imputation, but their performance degrades significantly in out-of-sample imputation. IGRM even fails on all datasets in the out-of-sample imputation setting. In contrast, our DIFFPUTER demonstrates excellent performance in both in-sample and out-of-sample imputation. This highlights DIFFPUTER's superior performance and robust generalization capabilities.

## 5.3 ABLATION STUDIES

**Impacts of the number of EM iterations.** In Fig. 3, we present the performance of DIFFPUTER's imputation results from increasing EM iterations. Note that $k = 0$ represents the imputation result of a randomly initialized denoising network, and $k = 1$ represents the performance of a pure diffusion model without iterative refinement. It is clearly observed that a single diffusion imputation achieves only suboptimal performance, while DIFFPUTER's performance steadily improves as the number of

Table 1: MCAR, In-sample imputation performance on **discrete columns**. Comparison of the imputation accuracy on discrete columns on five datasets. DIFFPUTER ranks the first among **19** imputation methods. Mean/Median/MF means using the mean/median/most frequent value as the imputation, which will give the same imputation result in this case.

| Method | Adult | Default | Shoppers | News | Average | Rank |
|---|---|---|---|---|---|---|
| *Statistical* | | | | | | |
| Mean/Median/MF | $55.20_{\pm0.02}$ | $53.37_{\pm0.11}$ | $50.60_{\pm0.29}$ | $18.60_{\pm0.14}$ | 44.44 | ⑮ |
| *Traditional iterative* | | | | | | |
| EM (García-Laencina et al., 2010) | $61.27_{\pm0.12}$ | $57.80_{\pm0.22}$ | $50.94_{\pm0.27}$ | $39.37_{\pm0.06}$ | 52.35 | ⑩ |
| MICE (Van Buuren & Karin, 2011) | $50.31_{\pm0.21}$ | $51.88_{\pm0.30}$ | $43.43_{\pm0.19}$ | $30.05_{\pm1.17}$ | 43.92 | ⑯ |
| MIRACLE (Kyono et al., 2021) | $62.28_{\pm0.21}$ | $55.79_{\pm1.52}$ | $45.74_{\pm0.69}$ | $39.33_{\pm0.27}$ | 50.79 | ⑫ |
| SoftImpute (Hastie et al., 2015) | $56.18_{\pm0.08}$ | $56.01_{\pm0.07}$ | $50.94_{\pm0.27}$ | $18.70_{\pm0.05}$ | 45.46 | ⑭ |
| MissForest (Stekhoven & Bühlmann, 2012) | $63.51_{\pm0.31}$ | $56.97_{\pm0.23}$ | $51.10_{\pm0.61}$ | $37.55_{\pm0.84}$ | 52.28 | ⑪ |
| *Dist. match* | | | | | | |
| MOT (Muzellec et al., 2020) | $63.85_{\pm0.16}$ | $71.95_{\pm0.06}$ | $56.91_{\pm0.11}$ | $38.62_{\pm0.20}$ | 57.83 | ⑤ |
| TDM (Zhao et al., 2023) | $62.64_{\pm1.36}$ | $63.41_{\pm7.16}$ | $55.00_{\pm0.46}$ | $30.62_{\pm1.22}$ | 52.92 | ⑧ |
| *GNN* | | | | | | |
| GRAPE (You et al., 2020) | $69.84_{\pm0.25}$ | $73.85_{\pm0.01}$ | $57.32_{\pm0.15}$ | $40.83_{\pm0.44}$ | 60.46 | ③ |
| IGRM (Zhong et al., 2023) | $69.21_{\pm0.29}$ | OOM | $57.45_{\pm0.16}$ | OOM | - | - |
| *Generative models* | | | | | | |
| MIWAE (Mattei & Frellsen, 2019) | $51.33_{\pm0.08}$ | $48.11_{\pm0.02}$ | $49.23_{\pm0.61}$ | $17.32_{\pm0.01}$ | 41.50 | ⑱ |
| GAIN (Yoon et al., 2018) | $53.73_{\pm0.38}$ | $54.49_{\pm1.80}$ | $47.25_{\pm0.95}$ | $34.20_{\pm0.06}$ | 47.42 | ⑬ |
| MCFlow (Richardson et al., 2020) | $60.31_{\pm0.18}$ | $68.36_{\pm0..9}$ | $52.73_{\pm0.29}$ | $30.08_{\pm3.01}$ | 52.87 | ⑨ |
| TabCSDI (Zheng & Charoenphakdee, 2022) | $58.72_{\pm0.25}$ | $50.57_{\pm0.02}$ | $43.24_{\pm0.05}$ | $17.49_{\pm0.14}$ | 42.51 | ⑰ |
| MissDiff (Ouyang et al., 2023) | $57.39_{\pm7.34}$ | $66.01_{\pm1.88}$ | $48.10_{\pm1.86}$ | $41.39_{\pm0.59}$ | 53.22 | ⑦ |
| *Other* | | | | | | |
| KNN (Pujianto et al., 2019) | $63.28_{\pm0.52}$ | $71.64_{\pm0.34}$ | $53.93_{\pm0.28}$ | $38.59_{\pm0.17}$ | 56.86 | ⑥ |
| Remasker (Du et al., 2024) | $69.18_{\pm0.08}$ | $76.88_{\pm0.62}$ | $57.86_{\pm0.09}$ | $44.33_{\pm0.08}$ | 62.06 | ② |
| HyperImpute (Jarrett et al., 2022) | $64.99_{\pm0.06}$ | $74.39_{\pm2.07}$ | $\mathbf{59.19}_{\pm0.04}$ | $40.29_{\pm0.96}$ | 59.72 | ④ |
| DIFFPUTER | $\mathbf{70.12}_{\pm0.17}$ | $\mathbf{77.64}_{\pm0.32}$ | $58.82_{\pm0.09}$ | $\mathbf{44.69}_{\pm0.13}$ | $\mathbf{62.82}$ | ① |

Figure 5: Impacts of sampling steps on the training time and imputation performance (MAE). Reducing diffusion sampling steps greatly reduces the training time at the cost of a slight performance drop.

Figure 6: Impacts of extremely large missing ratios. The performance of DIFFPUTER is upper-bounded by the initialized missing values via mean imputation

iterations increases. Additionally, we observe that DIFFPUTER does not require a large number of iterations to converge. In fact, 4 to 5 iterations are sufficient for DIFFPUTER to converge to a stable and satisfying state.

**Impacts of the number of sampled imputations per iteration.** To obtain the expected value of missing entries, we have to sample a number of examples from the conditional distribution (via the reverse diffusion process). Therefore, the number $N$ should have a huge impact on DIFFPUTER's performance. In Figure 4, we study the impacts of the number of samples on the imputation results. As expected, a very small $N$ leads to poor performance and large variance, while increasing the sampling number consistently improves the performance and reduces the variance. An optimal and stable performance can be achieved at $N \geq 10$. Since the sampling time is linear w.r.t. to the sample number, we use $N = 10$ by default.

**Impacts of the number of sampling steps in diffusion.** Another hyperparameter impacting the performance and training speed is the number of sampling steps $M$ in the reverse process of diffusion.

Table 2: Comparison of the real training time on an Nvidia RTX 4090 GPU. DIFFPUTER has a similar training cost to SOTA methods, yet brings from $8\%$ to $25\%$ performance improvement.

| Datasets | MOT | TDM | GRAPE | IGRM | HyperImpute | Remasker | DIFFPUTER |
|---|---|---|---|---|---|---|---|
| Time: **California** | $446.47s$ | $216.91s$ | $635.7s$ | $1267.5s$ | $1277.3s$ | $1320.1s$ | $1927.2s$ |
| Time: **Adult** | $396.68s$ | $514.56s$ | $2347.1s$ | $3865.1s$ | $1806.9s$ | $1902.4s$ | $2142.9s$ |
| DIFFPUTER**'s perf. improve.** | $21.47\%$ | $21.37\%$ | $25.94\%$ | $20.97\%$ | $8.44\%$ | $10.65\%$ | - |

By default, DIFFPUTER set $M = 50$, and we study the impact of reducing $M$ in Figure 5. As demonstrated, reducing the diffusion sampling steps can greatly reduce the training time at the cost of only a slight performance drop. More detailed, reducing the sampling step from 50 to 20 reduces about $25\%$ of training time cost, with only $3\%$ performance drop. Therefore, when time costs are high, and precision requirements are not stringent, the training speed can be further increased by reducing the sampling steps.

**Impacts of large missing ratios.** In Figure 6, we study the performance of DIFFPUTER with missing ratio from $30\%$ to $99\%$ (observed ratio from $70\%$ to $1\%$). Consistent with the intuition, when the observed ratio decreases (i.e., missing ratio increases), there is a significant increase in the imputation MAE score, indicating that the imputation performance becomes increasingly worse. However, we can observe a clear performance lower bound when the missing ratio approaches $99\%$, which is close to the performance of directly imputing using the mean of observed values. This is because, in DIFFPUTER's first E-step, each column's missing values are always initialized as the mean of the observed ones. Therefore, DIFFPUTER can still give a reasonable imputation result even if the missing ratio is extremely large.

**Comparison of Training Time.** Intuitively, the training of DIFFPUTER would be very time-consuming, as it requires iteratively alternating between the training and sampling processes of the diffusion model. However, in practice, DIFFPUTER's training efficiency is significantly improved through the following design: 1) light-weighted MLP-architectured denoising function; 2) reduced number of sampling steps ($M = 50 \ll 1000$ in traditional diffusion methods). To demonstrate this, we compare the training time of DIFFPUTER with other SOTA imputation methods. As demonstrated in Table 2, DIFFPUTER has similar training cost (similar scale) to these representative SOTA methods, yet brings from $8\%$ to $25\%$ performance improvement, which is acceptable.

**Incorporating EM with other DGMs.** Finally, we try to combine EM algorithms with other deep generative models that are easy to perform conditional sampling, including MIWAE (Mattei & Frellsen, 2019), HIWAE (Nazabal et al., 2020a), VAEM (Ma et al., 2020) and HH-VAEM (Peis et al., 2022a). Based on Table 3, our experimental results demonstrate that combining EM with other Deep Generative Models leads to performance improvements. However, it's noteworthy that despite these improvements, our proposed DIFFPUTER still outperforms all combinations, achieving superior results, where lower scores indicate

Table 3: Effects of combining EM with other Deep Generative Models.

| Methods | Adult | Default | Shoppers | News |
|---|---|---|---|---|
| MIWAE | 0.5763 | 0.5194 | 0.5047 | 0.6349 |
| HIWAE | 0.6155 | 0.3989 | 0.4707 | 0.5032 |
| VAEM | 0.5568 | 0.4292 | 0.4626 | 0.5204 |
| HH-VAEM | 0.5673 | - | 0.4589 | - |
| EM + MIWAE | 0.5661 | 0.4993 | 0.4554 | 0.4171 |
| EM + HIWAE | 0.5974 | 0.4314 | 0.4961 | 0.5222 |
| EM + VAEM | 0.5492 | 0.4121 | 0.4362 | 0.5045 |
| EM + HH-VAEM | 0.5402 | - | 0.4262 | - |
| DIFFPUTER | **0.3425** | **0.2661** | **0.3485** | **0.2855** |

better performance. This demonstrates the effectiveness and robustness of our approach compared to existing methods, even when they are enhanced through combination with EM.

## 6 CONCLUSIONS

In this paper, we have proposed DIFFPUTER for missing data imputation. DIFFPUTER is an iterative method that combines the Expectation-Maximization algorithm and diffusion models, where the diffusion model serves as both the density estimator and missing data imputer. We demonstrate theoretically that the training and sampling process of a diffusion model precisely corresponds to the M-step and E-step of the EM algorithm. Therefore, we can iteratively update the density of the complete data and the values of the missing data. Extensive experiments have demonstrated the efficacy of the proposed method.

ACKNOWLEDGMENTS

This work is supported in part by NSF under grants III-2106758, and POSE-2346158.

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

## A  Definition of Symbols

In Table 4, we explain the definition of every symbol used in this paper.

Table 4: Explanations of the symbols used in this paper.

| Symbol | Explanation |
|---|---|
| $d$ | the number of columns of tabular data |
| $\mathbf{x} \in \mathbb{R}^d$ | random variable, representing a row of the tabular data |
| $p(\mathbf{x}) = p_{\text{data}}(\mathbf{x})$ | the distribution of the complete data |
| $\mathbf{m} \in \{0,1\}^d$ | binary mask vector |
| $\mathbf{x}^{\text{obs}}$ | the observed part of $\mathbf{x}$, indicated by $m_k = 0$ |
| $\mathbf{x}^{\text{mis}}$ | the missing part of $\mathbf{x}$, indicated by $m_k = 1$ |
| $t$ | the timestep of diffusion process |
| $\sigma(t) = t$ | the noise level at timestep $t$ |
| $T$ | the maximum timestep |
| $\varepsilon \sim \mathcal{N}(\mathbf{0}, \mathbf{I})$ | standard Gaussian noise |
| $\mathbf{x}_0 = \mathbf{x}$ | unforwarded $\mathbf{x}$ (at timestep 0) |
| $\mathbf{x}_0^{\text{obs}} = \mathbf{x}^{\text{obs}}$ | observed part of $\mathbf{x}$ |
| $\mathbf{x}_0^{\text{mis}} = \mathbf{x}^{\text{mis}}$ | missing part of $\mathbf{x}$ |
| $\mathbf{x}_t = \mathbf{x}_0 + \sigma(t)\varepsilon$ | forwarded $\mathbf{x}$ at timestep $t$ |
| $\mathbf{x}_t^{\text{obs}} = \mathbf{x}_0^{\text{obs}} + \sigma(t)\varepsilon$ | observed part of $\mathbf{x}_t$ |
| $\mathbf{x}_t^{\text{mis}} = \mathbf{x}_0^{\text{mis}} + \sigma(t)\varepsilon$ | missing part of $\mathbf{x}_t$ |
| $\theta$ | diffusion model's parameters |
| $p_\theta(\mathbf{x})$ | density of data induced by the diffusion model |
| $\tilde{\mathbf{x}}_T \sim \pi(\mathbf{x}) = \mathcal{N}(\mathbf{0}, \sigma^2(T)\mathbf{I})$ | sampled random noise for the reverse process |
| $\tilde{\mathbf{x}}_T^{\text{obs}}$ | observed part of $\tilde{\mathbf{x}}_T$ |
| $\tilde{\mathbf{x}}_T^{\text{mis}}$ | missing part of $\tilde{\mathbf{x}}_T$ |
| $\tilde{\mathbf{x}}_t$ | sampled data at timestep $t$ in the reverse process |
| $\tilde{\mathbf{x}}_t^{\text{obs}}$ | observed part of $\tilde{\mathbf{x}}_t$ |
| $\tilde{\mathbf{x}}_t^{\text{mis}}$ | missing part of $\tilde{\mathbf{x}}_t$ |
| $\hat{\mathbf{x}}_0^{\text{obs}}$ | The exact ground-truth value(s) of the observed part |

## B  Proofs

### B.1  Proof for Theorem 1

*Proof.* First, it is obvious that the observed entries from Eq. 7, i.e., $\tilde{\mathbf{x}}_t^{\text{obs}}$, satisfy

$$\tilde{\mathbf{x}}_t^{\text{obs}} \sim p(\mathbf{x}_t^{\text{obs}} | \mathbf{x}^{\text{obs}} = \hat{\mathbf{x}}_0^{\text{obs}}), \ \forall t. \tag{9}$$

Then, we introduce the following Lemma.

**Lemma 1.** *Let $\tilde{\mathbf{x}}_t \sim p_\theta(\mathbf{x}_t | \mathbf{x}^{\text{obs}} = \hat{\mathbf{x}}_0^{\text{obs}})$. If $\tilde{\mathbf{x}}_{t-\Delta t}$ is a sample obtained from Eq. 5, Eq. 6, and Eq. 7, then $\tilde{\mathbf{x}}_{t-\Delta t} \sim p_\theta(\mathbf{x}_{t-\Delta t} | \mathbf{x}^{\text{obs}} = \hat{\mathbf{x}}_0^{\text{obs}})$, when $\Delta t \to 0^+$.*

*Proof.* First, note that $\mathbf{x}_t$ ($\mathbf{x}_{t-\Delta t}$) is obtained via adding **dimensionally-independent** Gaussian noises to $\mathbf{x}_0 = \mathbf{x}$ (see the forward process in Eq. 1), we have

$$
\begin{aligned}
p_\theta(\mathbf{x}_{t-\Delta t} | \mathbf{x}^{\text{obs}} = \hat{\mathbf{x}}_0^{\text{obs}}) &= p_\theta(\mathbf{x}_{t-\Delta t}^{\text{obs}}, \mathbf{x}_{t-\Delta t}^{\text{mis}} | \mathbf{x}^{\text{obs}} = \hat{\mathbf{x}}_0^{\text{obs}}) \\
&= \underbrace{p(\mathbf{x}_{t-\Delta t}^{\text{obs}} | \mathbf{x}^{\text{obs}} = \hat{\mathbf{x}}_0^{\text{obs}})}_{\text{Correspond to } \tilde{\mathbf{x}}_{t-\Delta t}^{\text{obs}} \text{ in Eq. 5}} \cdot p_\theta(\mathbf{x}_{t-\Delta t}^{\text{mis}} | \mathbf{x}^{\text{obs}} = \hat{\mathbf{x}}_0^{\text{obs}}).
\end{aligned}
\tag{10}
$$

Therefore, $\tilde{\mathbf{x}}_{t-\Delta t}$'s observed entries $\tilde{\mathbf{x}}_{t-\Delta t}^{\text{obs}}$ is sampled from distribution $p(\mathbf{x}_{t-\Delta t}^{\text{obs}}|\mathbf{x}^{\text{obs}} = \hat{\mathbf{x}}_0^{\text{obs}})$. Then, we turn to the missing entries, where we have the following derivation:

$$
\begin{aligned}
p_{\boldsymbol{\theta}}(\mathbf{x}_{t-\Delta t}^{\text{mis}}|\mathbf{x}^{\text{obs}} = \hat{\mathbf{x}}_0^{\text{obs}}) &= \int p_{\boldsymbol{\theta}}(\mathbf{x}_{t-\Delta t}^{\text{mis}}|\mathbf{x}_t, \mathbf{x}^{\text{obs}} = \hat{\mathbf{x}}_0^{\text{obs}})p_{\boldsymbol{\theta}}(\mathbf{x}_t|\mathbf{x}^{\text{obs}} = \hat{\mathbf{x}}_0^{\text{obs}})\mathrm{d}\mathbf{x}_t \\
&= \mathbb{E}_{p_{\boldsymbol{\theta}}(\mathbf{x}_t|\mathbf{x}^{\text{obs}}=\hat{\mathbf{x}}_0^{\text{obs}})}p_{\boldsymbol{\theta}}(\mathbf{x}_{t-\Delta t}^{\text{mis}}|\mathbf{x}_t, \mathbf{x}^{\text{obs}} = \hat{\mathbf{x}}_0^{\text{obs}}) \\
&\approx \mathbb{E}_{p_{\boldsymbol{\theta}}(\mathbf{x}_t|\mathbf{x}^{\text{obs}}=\hat{\mathbf{x}}_0^{\text{obs}})}p_{\boldsymbol{\theta}}(\mathbf{x}_{t-\Delta t}^{\text{mis}}|\mathbf{x}_t) \\
&= p_{\boldsymbol{\theta}}(\mathbf{x}_{t-\Delta t}^{\text{mis}}|\tilde{\mathbf{x}}_t),
\end{aligned}
\tag{11}
$$

where $\tilde{\mathbf{x}}_t$ is a random sample from $p_{\boldsymbol{\theta}}(\mathbf{x}_t|\mathbf{x}^{\text{obs}} = \hat{\mathbf{x}}_0^{\text{obs}})$. The first '$\approx$' holds because when $\Delta t \to 0$, $\mathbf{x}_{t-\Delta t}^{\text{mis}}$ is almost predictable via $\mathbf{x}_t$ without $\mathbf{x}_0^{\text{obs}}$. Therefore, when $\tilde{\mathbf{x}}_t \sim p_{\boldsymbol{\theta}}(\mathbf{x}_t|\mathbf{x}^{\text{obs}} = \hat{\mathbf{x}}_0^{\text{obs}})$, $p_{\boldsymbol{\theta}}(\mathbf{x}_{t-\Delta t}^{\text{mis}}|\mathbf{x}^{\text{obs}} = \hat{\mathbf{x}}_0^{\text{obs}})$ is approximately tractable via $p_{\boldsymbol{\theta}}(\mathbf{x}_{t-\Delta t}^{\text{mis}}|\tilde{\mathbf{x}}_t)$.

Note that $\mathbf{x}_{t-\Delta t}^{\text{mis}}$ denotes the missing entries of $\mathbf{x}_{t-\Delta t}$. According to the reverse denoising process in Eq. 2, given $\mathbf{x}_t = \tilde{\mathbf{x}}_t$, $\mathbf{x}_{t-\Delta t}$ is obtained via integrating $\mathrm{d}\mathbf{x}_t$ from $t$ to $t - \Delta t$, i.e.,

$$
\mathbf{x}_{t-\Delta t} = \tilde{\mathbf{x}}_t + \int_t^{t-\Delta t} \mathrm{d}\mathbf{x}_t.
\tag{12}
$$

Therefore, $\tilde{\mathbf{x}}_{t-\Delta t}^{\text{mis}} = \mathbf{x}_{t-\Delta t}^{\text{mis}}$ obtained from Eq. 6 is a sample from $p_{\boldsymbol{\theta}}(\mathbf{x}_{t-\Delta t}^{\text{mis}}|\tilde{\mathbf{x}}_t)$. And, approximately, $\tilde{\mathbf{x}}_{t-\Delta t}^{\text{mis}} \sim p_{\boldsymbol{\theta}}(\mathbf{x}_{t-\Delta t}^{\text{mis}}|\mathbf{x}^{\text{obs}} = \hat{\mathbf{x}}_0^{\text{obs}})$.

Since $\tilde{\mathbf{x}}_{t-\Delta t}^{\text{obs}} \sim p(\mathbf{x}_{t-\Delta t}^{\text{obs}}|\mathbf{x}^{\text{obs}} = \hat{\mathbf{x}}_0^{\text{obs}}), \tilde{\mathbf{x}}_{t-\Delta t}^{\text{mis}} \sim p_{\boldsymbol{\theta}}(\mathbf{x}_{t-\Delta t}^{\text{mis}}|\mathbf{x}^{\text{obs}} = \hat{\mathbf{x}}_0^{\text{obs}})$, we have $\tilde{\mathbf{x}}_{t-\Delta t} \sim p_{\boldsymbol{\theta}}(\mathbf{x}_{t-\Delta t}|\mathbf{x}^{\text{obs}} = \hat{\mathbf{x}}_0^{\text{obs}})$, according to Eq. 10. Therefore, the proof for Lemma 1 is completed. □

With Lemma 1, we are able to prove Theorem 1 via induction, as long as $\tilde{\mathbf{x}}_T$ is also sampled from $p(\mathbf{x}_T|\mathbf{x}^{\text{obs}} = \hat{\mathbf{x}}_0^{\text{obs}})$. This holds because $p(\mathbf{x}_T|\mathbf{x}_0) \approx p(\mathbf{x}_T)$, given the condition that $\mathbf{x}_0 = \mathbf{x}$ has zero mean and unit variance, and $\sigma(T) \gg 1$.

Note that the score function $\nabla_{\mathbf{x}_t} \log p(\mathbf{x}_t)$ in Eq. 2 is intractable, it is replaced with the output of the score neural network $\boldsymbol{\epsilon}_{\boldsymbol{\theta}}(\mathbf{x}_t, t)$. Therefore, the finally obtained distribution can be rewritten as $p_{\boldsymbol{\theta}}(\mathbf{x}|\mathbf{x}^{\text{obs}} = \hat{\mathbf{x}}_0^{\text{obs}})$. Therefore, the proof for Theorem 1 is complete.

□

## C  FURTHER EXPLANATION OF THE DIFFUSION MODEL

The diffusion model we adopt in Section 4.1 is actually a simplified version of the Variance-Exploding SDE proposed in (Song et al., 2021b).

Note that (Song et al., 2021b) has provided a unified formulation via the Stochastic Differential Equation (SDE) and defines the forward process of Diffusion as

$$
\mathrm{d}\mathbf{x} = \boldsymbol{f}(\mathbf{x}, t)\mathrm{d}t + g(t)\,\mathrm{d}\boldsymbol{w}_t,
\tag{13}
$$

where $\boldsymbol{f}(\cdot)$ and $g(\cdot)$ are the drift and diffusion coefficients and are selected differently for different diffusion processes, e.g., the variance preserving (VP) and variance exploding (VE) formulations. $\boldsymbol{\omega}_t$ is the standard Wiener process. Usually, $\boldsymbol{f}(\cdot)$ is of the form $\boldsymbol{f}(\mathbf{x}, t) = f(t)\,\mathbf{x}$. Thus, the SDE can be equivalently written as

$$
\mathrm{d}\mathbf{x} = f(t)\,\mathbf{x}\,\mathrm{d}t + g(t)\,\mathrm{d}\boldsymbol{w}_t.
\tag{14}
$$

Let $\mathbf{x}$ be a function of the time $t$, i.e., $\mathbf{x}_t = \mathbf{x}(t)$, then the conditional distribution of $\mathbf{x}_t$ given $\mathbf{x}_0$ (named as the perturbation kernel of the SDE) could be formulated as:

$$
p(\mathbf{x}_t|\mathbf{x}_0) = \mathcal{N}(\mathbf{x}_t; s(t)\mathbf{x}_0, s^2(t)\sigma^2(t)\boldsymbol{I}),
\tag{15}
$$

where

$$
s(t) = \exp\left(\int_0^t f(\xi)\mathrm{d}\xi\right), \text{ and } \sigma(t) = \sqrt{\int_0^t \frac{g^2(\xi)}{s^2(\xi)}\mathrm{d}\xi}.
\tag{16}
$$

Therefore, the forward diffusion process could be equivalently formulated by defining the perturbation kernels (via defining appropriate $s(t)$ and $\sigma(t)$).

Variance Exploding (VE) implements the perturbation kernel Eq. 15 by setting $s(t) = 1$, indicating that the noise is directly added to the data rather than weighted mixing. Therefore, The noise variance (the noise level) is totally decided by $\sigma(t)$. The diffusion model used in DIFFPUTER belongs to VE-SDE, but we use linear noise level (i.e., $\sigma(t) = t$) rather than $\sigma(t) = \sqrt{t}$ in the vanilla VE-SDE (Song et al., 2021b). When $s(t) = 1$, the perturbation kernels become:

$$p(\mathbf{x}_t|\mathbf{x}_0) = \mathcal{N}(\mathbf{x}_t; \mathbf{0}, \sigma^2(t)\mathbf{I}) \;\Rightarrow\; \mathbf{x}_t = \mathbf{x}_0 + \sigma(t)\boldsymbol{\varepsilon}, \tag{17}$$

which aligns with the forward diffusion process in Eq. 1.

The sampling process of diffusion SDE is given by:

$$\mathrm{d}\mathbf{x} = [\boldsymbol{f}(\mathbf{x}, t) - g^2(t)\nabla_{\mathbf{x}} \log p_t(\mathbf{x})]\mathrm{d}t + g(t)\mathrm{d}\boldsymbol{w}_t. \tag{18}$$

For VE-SDE, $s(t) = 1 \Leftrightarrow \boldsymbol{f}(\mathbf{x}, t) = f(t) \cdot \mathbf{x} = \mathbf{0}$, and

$$
\begin{aligned}
\sigma(t) &= \sqrt{\int_0^t g^2(\xi)\mathrm{d}\xi} \Rightarrow \int_0^t g^2(\xi)\mathrm{d}\xi = \sigma^2(t), \\
g^2(t) &= \frac{\mathrm{d}\sigma^2(t)}{\mathrm{d}t} = 2\sigma(t)\dot{\sigma}(t), \\
g(t) &= \sqrt{2\sigma(t)\dot{\sigma}(t)}.
\end{aligned}
\tag{19}
$$

Plugging $g(t)$ into Eq. 18, the reverse process in Eq. 2 is recovered:

$$\mathrm{d}\mathbf{x}_t = -2\sigma(t)\dot{\sigma}(t)\nabla_{\mathbf{x}_t} \log p(\mathbf{x}_t)\mathrm{d}t + \sqrt{2\sigma(t)\dot{\sigma}(t)}\mathrm{d}\boldsymbol{\omega}_t. \tag{20}$$

## D  EXPERIMENTAL DETAILS

### D.1  CONFIGURATIONS.

We conduct all experiments with:

- Operating System: Ubuntu 22.04.3 LTS
- CPU: Intel 13th Gen Intel(R) Core(TM) i9-13900K
- GPU: NVIDIA GeForce RTX 4090 with 24 GB of Memory
- Software: CUDA 12.2, Python 3.9.16, PyTorch (Paszke et al., 2019) 1.12.1

### D.2  DATASETS

We use ten real-world datasets of varying scales, and all of them are available at Kaggle[2] or the UCI Machine Learning repository[3]. We consider five datasets of only continuous features: California[4], Letter[5], Gestur[6], Magic[7], and Bean[8], and five datasets of both continuous and discrete features: Adult[9], Default[10], Shoppers[11], and News[12]. The statistics of these datasets are presented in Table 5.

---

[2] https://www.kaggle.com/
[3] https://archive.ics.uci.edu/
[4] https://www.kaggle.com/datasets/camnugent/california-housing-prices
[5] https://archive.ics.uci.edu/dataset/59/letter+recognition
[6] https://archive.ics.uci.edu/dataset/302/gesture+phase+segmentation
[7] https://archive.ics.uci.edu/dataset/159/magic+gamma+telescope
[8] https://archive.ics.uci.edu/dataset/602/dry+bean+dataset
[9] https://archive.ics.uci.edu/dataset/2/adult
[10] https://archive.ics.uci.edu/dataset/350/default+of+credit+card+clients
[11] https://archive.ics.uci.edu/dataset/468/online+shoppers+purchasing+intention+dataset
[12] https://archive.ics.uci.edu/dataset/332/online+news+popularity

Table 5: Statistics of datasets. # Num stands for the number of numerical columns, and # Cat stands for the number of categorical columns.

| Dataset | # Rows | # Num | # Cat | # Train (In-sample) | # Test (Out-of-Sample) |
|---|---|---|---|---|---|
| **California** Housing | 20,640 | 9 | - | 14,303 | 6,337 |
| **Letter** Recognition | 20,000 | 16 | - | 14,000 | 6,000 |
| **Gesture** Phase Segmentation | 9,522 | 32 | - | 6,665 | 2,857 |
| **Magic** Gamma Telescope | 19,020 | 10 | - | 13,314 | 5,706 |
| Dry **Bean** | 13,610 | 17 | - | 9,527 | 4,083 |
| **Adult** Income | 32,561 | 6 | 8 | 22,792 | 9,769 |
| **Default** of Credit Card Clients | 30,000 | 14 | 10 | 21,000 | 9,000 |
| Online **Shoppers** Purchase | 12,330 | 10 | 7 | 8,631 | 3,699 |
| Online **News** Popularity | 39,644 | 45 | 2 | 27,790 | 11,894 |

## D.3 MISSING MECHANISMS

According to how the masks $\mathbf{m}$ are generated, there are three mainstream mechanisms of missingness, namely missing patterns: 1) Missing completely at random (MCAR) refers to the case that the probability of an entry being missing is independent of the data, i.e., $p(\mathbf{m}|\mathbf{x}) = p(\mathbf{m})$. 2) In missing at random (MAR), the probability of missingness depends only on the observed values, i.e., $p(\mathbf{m}|\mathbf{x}) = p(\mathbf{m}|\mathbf{x}^{\text{obs}})$ 3) All other cases are classified as missing not at random (MNAR), where the probability of missingness might also depend on other missing entries.

We follow the methods proposed in Zhao et al. (2023) to implement MAR and MNAR.

- "For MAR, we first sample a subset of features (columns in $X$) that will not contain missing values, and then we use a logistic model with these non-missing columns as input to determine the missing values of the remaining columns, and we employ line search of the bias term to get the desired proportion of missing values."

- For MNAR, we use the first approach proposed in [4], "Using a logistic model with the input masked by MCAR". Specifically, similar to the MAR setting, we first divide the data columns into two groups, with one group serving as input for the logistic model, outputting the missing probability for the other set of columns. The difference is that after determining the missing probability of the second set, we apply MCAR to the input columns (the first set). Hence, missing values from the second set will depend on the masked values of the first set.

The code for generating masks according to the three missing mechanisms is also provided at https://anonymous.4open.science/r/ICLR-DiffPuter.

## D.4 IMPLEMENTATIONS AND HYPERPARAMETERS

We use a fixed set of hyperparameters, which will save significant efforts in hyperparameter-tuning when applying DIFFPUTER to more datasets. For the diffusion model, we set the maximum time $T = 80$, the noise level $\sigma(t) = t$, which is linear to $t$. The score/denoising neural network $\boldsymbol{\epsilon}(\mathbf{x}_t, t)$ is implemented as a 5-layer MLP with hidden dimension 1024. $t$ is transformed to sinusoidal timestep embeddings and then added to $\mathbf{x}_t$, which is subsequently passed to the denoising function. When using the learned diffusion model for imputation, we set the number of discrete steps $M = 50$ and the number of sampling times per data sample $N = 10$. DIFFPUTER is implemented with Pytorch, and optimized using Adam (Kingma & Ba, 2015) optimizer with a learning rate of $1 \times 10^{-4}$.

**Architecture of denoising neural network.** We use the same architecture of denoising neural network as in two recent tabular diffusion models for tabular data synthesis (Kotelnikov et al., 2023; Zhang et al., 2024). The denoising MLP takes the current time step $t$ and the feature vector $\mathbf{x}_t \in \mathbb{R}^{1 \times d}$ as input. First, $\mathbf{x}_t$ is fed into a linear projection layer that converts the vector dimension to be $d_{\text{hidden}} = 1024$:

$$\mathbf{h}_0 = \text{FC}_{\text{in}}(\mathbf{x}_t) \in \mathbb{R}^{1 \times d_{\text{hidden}}}, \tag{21}$$

where $\mathbf{h}_0$ is the transformed vector, and $d_{\text{hidden}}$ is the output dimension of the input layer.

Then, following the practice in TabDDPM (Kotelnikov et al., 2023), the sinusoidal timestep embeddings $\mathbf{t}_{\mathrm{emb}} \in \mathbb{R}^{1 \times d_{\mathrm{hidden}}}$ is added to $\mathbf{h}_0$ to obtain the input vector $\mathbf{h}_{\mathrm{hidden}}$:

$$\mathbf{h}_{\mathrm{in}} = \mathbf{h}_0 + \mathbf{t}_{\mathrm{emb}}. \tag{22}$$

The hidden layers are three fully connected layers of the size $d_{\mathrm{hidden}} - 2*d_{\mathrm{hidden}} - 2*d_{\mathrm{hidden}} - d_{\mathrm{hidden}}$, with SiLU activation functions:

$$\begin{aligned}
\mathbf{h}_1 &= \mathtt{SiLU}(\mathtt{FC}_1(\mathbf{h}_0) \in \mathbb{R}^{1 \times 2*d_{\mathrm{hidden}}}), \\
\mathbf{h}_2 &= \mathtt{SiLU}(\mathtt{FC}_2(\mathbf{h}_1) \in \mathbb{R}^{1 \times 2*d_{\mathrm{hidden}}}), \\
\mathbf{h}_3 &= \mathtt{SiLU}(\mathtt{FC}_3(\mathbf{h}_2) \in \mathbb{R}^{1 \times d_{\mathrm{hidden}}}).
\end{aligned} \tag{23}$$

The estimated score is obtained via the last linear layer:

$$\boldsymbol{\epsilon}_\theta(\mathbf{x}_t, t) = \mathbf{h}_{\mathrm{out}} = \mathtt{FC}_{\mathrm{out}}(\mathbf{h}_3) \in \mathbb{R}^{1 \times d}. \tag{24}$$

Finally, $\boldsymbol{\epsilon}_\theta(\mathbf{x}_t, t)$ is applied to Eq. 3 for model training.

### D.5 IMPLEMENTATIONS OF BASELINES

We implement most of the baseline methods according to the publicly available codebases:

- Remasker (Du et al., 2024): `https://github.com/tydusky/remasker`.
- TDM (Zhao et al., 2023): `https://github.com/hezgit/TDM`
- MOT (Muzellec et al., 2020): `https://github.com/BorisMuzellec/MissingDataOT`
- GRAPE (You et al., 2020): `https://github.com/maxiaoba/GRAPE`
- IGRM (Zhong et al., 2023): `https://github.com/G-AILab/IGRM`
- TabCSDI (Zheng & Charoenphakdee, 2022): `https://github.com/pfnet-research/TabCSDI`
- MCFlow (Richardson et al., 2020): `https://github.com/trevor-richardson/MCFlow`.
- For HyperImputer (Jarrett et al., 2022), MissForest (Stekhoven & Bühlmann, 2012), MICE (Van Buuren & Karin, 2011), SoftImpute (Hastie et al., 2015), EM (García-Laencina et al., 2010), GAIN (Yoon et al., 2018), MIRACLE (Kyono et al., 2021), and MIWAE (Mattei & Frellsen, 2019), we use the implementations at: `https://github.com/vanderschaarlab/hyperimpute`.

MissDiff (Ouyang et al., 2023) does not provide its official implementations. Therefore, we obtain its results based on our own implementation.

The codes for all the methods are available at `https://anonymous.4open.science/r/ICLR-DiffPuter`

### D.6 HYPERPARAMETER SETTINGS OF BASELINES

Most of the deep learning baselines recommend the use of one set of hyperparameters for all datasets. For these methods, we directly follow their guidelines and use the default hyperparameters:

- ReMasker (Du et al., 2024): we use the recommended hyperparameters provided in Appendix A.2 in the original paper (Du et al., 2024). This set of hyperparameters is searched by tuning on the Letter dataset and is deployed for all the datasets in the original paper; hence, we follow this setting.
- HyperImpute (Jarrett et al., 2022): since HyperImpute works by searching over the space of classifiers/regressors and their hyperparameters, it does not have hyperparameters itself except parameters related to the AutoML search budget. We adopt the default budget parameters of HyperImpute's official implementation for all datasets. The default budget

    parameters and AutoML search space are provided[13] and Table 5 in the original paper (Jarrett et al., 2022).

- MOT (Muzellec et al., 2020) and TDM (Zhao et al., 2023): There is a main hyperparameter representing the number of subset pairs sampled from the dataset for computing optimal transport loss. Sinkhorn algorithm and TDM are controlled by hyperparameter n_iter. While the default value is 3000, we set it as 12000 for all datasets to ensure the algorithm converges sufficiently. For the round-robin version of the algorithm, the number of sampled pairs is controlled by max_iter and rr_iter; we adopt the default value 15, which is enough for the algorithm to converge. For the remaining hyperparameters related to network architectures, we use the default ones for all datasets.

- kNN: we follow the common practice of selecting the number of nearest neighbors as $\sqrt{n}$, where $n$ is the number of samples in the dataset.

- GRAPE (You et al., 2020) and IGRM (Zhong et al., 2023): we adopt the recommended set of hyperparameters used in the original paper for all datasets. For a detailed explanation of the meaning of the parameters, please see the github repos: GRAPE[14] and IGRM[15].

- MissDiff: since the original implementation is not available and it is based on the diffusion model, for a fair comparison, we simply use the same set of hyperparameters with our DiffPuter.

- TabCSDI (Zheng & Charoenphakdee, 2022): we follow the guide for selecting hyperparameters in the original paper (Appendix B in [3]). Specifically, we use a large version of the TabCSDI model with a number of layers set to 4 (see more detailed hyperparameters about the large TabCSDI model[16]. For batch size, we take the official choice of batch size (8) for the breast dataset ( 700 samples) as a base and scale the batch size accordingly with the sample size of our datasets: since most of the datasets we used have the number of samples between 20000 to 40000, we scale the batch size to 256 and use it for all datasets.

- MCFlow (Richardson et al., 2020): we adopt the recommended hyperparameters provided in the official implementation for all datasets[17].

- For the remaining classical machine learning methods, including EM, GAIN, MICE, Miracle, MissForest, and Softimpute where hyperparameters might be important. Since we use the implementations from the 'hyperimpute' package, we tune the hyperparameters within the hyperparameter space provided in the package[18]. To be specific, we set the maximum budge as 50, then we sample 50 different hyperparameter combinations according to the hyperparameter space. Finally, we report the optimal performance over the 50 trials.

# E    ADDITIONAL RESULTS

## E.1    MCAR: OUT-OF-SAMPLE IMPUTATION

We present the out-of-sample imputation performance comparison under MCAR setting in Table 6.

## E.2    MISSING AT RANDOM (MAR)

We present the performance comparison under MAR setting in Table 7 and Table 8.

## E.3    MISSING NOT AT RANDOM (MNAR)

We present the performance comparison under MNAR setting in Table 9 and Table 10.

---

[13]https://github.com/vanderschaarlab/hyperimpute/blob/main/src/hyperimpute/plugins/imputers/plugin_hyperimpute.py

[14]https://github.com/maxiaoba/GRAPE/blob/master/train_mdi.py

[15]https://github.com/G-AILab/IGRM/blob/main/main.py

[16]https://github.com/pfnet-research/TabCSDI/blob/main/config/census_onehot_analog.yaml

[17]https://github.com/trevor-richardson/MCFlow/blob/master/main.py

[18]https://github.com/vanderschaarlab/hyperimpute/tree/main/src/hyperimpute/plugins/imputers/plugin_missforest.py

Table 6: MCAR, Out-of-sample imputation performance on MAE and RMSE metrics (using base $10^{-2}$ for better presentation). DIFFPUTER outperforms the most competitive baseline methods by $13.37\%$ on MAE, and by $4.43\%$ on RMSE.

| | Method | California | Letter | Gesture | Adult | Default | Shoppers | Average |
|---|---|---|---|---|---|---|---|---|
| **MAE** | MCFlow | $60.74_{\pm 5.99}$ | $66.35_{\pm 0.62}$ | $82.72_{\pm 2.43}$ | $77.54_{\pm 0.98}$ | $64.12_{\pm 2.16}$ | $73.90_{\pm 3.34}$ | 70.90 |
| | IGRM | $138.22_{\pm 4.25}$ | $121.59_{\pm 1.16}$ | $149.88_{\pm 3.92}$ | $149.64_{\pm 0.18}$ | OOM | $132.28_{\pm 0.01}$ | - |
| | GRAPE | $36.54_{\pm 0.12}$ | $42.41_{\pm 0.19}$ | $92.50_{\pm 0.07}$ | $59.25_{\pm 0.58}$ | $57.51_{\pm 0.09}$ | $51.94_{\pm 1.20}$ | 56.69 |
| | MOT | $44.20_{\pm 0.81}$ | $46.75_{\pm 0.13}$ | $36.74_{\pm 0.01}$ | $51.17_{\pm 0.32}$ | $31.42_{\pm 0.50}$ | $41.55_{\pm 0.42}$ | 41.97 |
| | Remasker | $33.97_{\pm 0.40}$ | $34.03_{\pm 0.33}$ | $35.29_{\pm 4.13}$ | $49.27_{\pm 0.53}$ | $37.02_{\pm 3.38}$ | $41.59_{\pm 1.39}$ | 38.53 |
| | **DIFFPUTER** | $\mathbf{33.47}_{\pm 0.25}$ | $\mathbf{30.69}_{\pm 0.61}$ | $\mathbf{28.53}_{\pm 0.18}$ | $\mathbf{49.14}_{\pm 0.62}$ | $\mathbf{24.63}_{\pm 0.24}$ | $\mathbf{34.46}_{\pm 0.48}$ | **33.49** |
| | Improv. | **1.47**% | **9.81**% | **19.16**% | **0.26**% | **21.61**% | **17.06**% | **13.09**% |
| **RMSE** | MCFlow | $84.57_{\pm 7.57}$ | $87.73_{\pm 0.58}$ | $119.73_{\pm 1.10}$ | $120.68_{\pm 0.66}$ | $101.33_{\pm 1.82}$ | $115.75_{\pm 0.82}$ | 104.97 |
| | IGRM | $170.00_{\pm 4.72}$ | $151.16_{\pm 2.05}$ | $172.25_{\pm 2.99}$ | $179.17_{\pm 0.88}$ | OOM | $174.42_{\pm 1.73}$ | - |
| | GRAPE | $56.05_{\pm 0.03}$ | $58.11_{\pm 0.29}$ | $120.31_{\pm 0.11}$ | $99.19_{\pm 0.50}$ | $91.94_{\pm 0.42}$ | $92.37_{\pm 0.41}$ | 86.33 |
| | MOT | $71.02_{\pm 1.19}$ | $64.69_{\pm 0.44}$ | $71.64_{\pm 0.84}$ | $94.67_{\pm 1.50}$ | $71.35_{\pm 1.08}$ | $82.63_{\pm 2.19}$ | 76 |
| | Remasker | $\mathbf{52.96}_{\pm 0.09}$ | $48.60_{\pm 0.45}$ | $68.78_{\pm 3.60}$ | $\mathbf{90.38}_{\pm 1.41}$ | $76.94_{\pm 1.72}$ | $80.00_{\pm 0.88}$ | 69.61 |
| | **DIFFPUTER** | $54.18_{\pm 0.25}$ | $\mathbf{47.74}_{\pm 0.39}$ | $\mathbf{60.90}_{\pm 0.52}$ | $91.31_{\pm 0.61}$ | $\mathbf{70.50}_{\pm 0.83}$ | $\mathbf{74.01}_{\pm 0.40}$ | **66.41** |
| | Improv. | - | **1.77**% | **11.46**% | - | **1.19**% | **7.49**% | **4.60**% |

Table 7: MAR, In-sample imputation MAE.

| Method | California | Letter | Gesture | Magic | Bean | Adult | Default | Shoppers | News |
|---|---|---|---|---|---|---|---|---|---|
| *Traditional iterative* | | | | | | | | | |
| EM | 39.17 | 58.05 | 35.43 | 48.13 | 15.41 | 62.54 | 33.78 | 43.01 | 39.52 |
| MICE | 58.56 | 82.32 | 65.14 | 72.81 | 22.17 | 98.44 | 63.92 | 75.70 | 61.12 |
| MIRACLE | 47.13 | 71.10 | 62.56 | 53.87 | 29.06 | 73.06 | 43.72 | 56.89 | 38.21 |
| SoftImpute | 59.47 | 67.01 | 53.95 | 58.55 | 31.55 | 70.39 | 42.66 | 61.91 | 64.85 |
| MissForest | 45.90 | 62.51 | 37.69 | 47.70 | 29.21 | 73.25 | 38.09 | 41.97 | 41.18 |
| *Dist. match* | | | | | | | | | |
| MOT | 44.52 | 50.86 | 44.38 | 46.95 | 30.41 | 58.00 | 36.48 | 40.53 | 49.92 |
| TDM | 39.27 | 46.86 | 42.69 | 45.31 | 29.11 | 59.59 | 37.93 | 45.00 | 63.25 |
| *GNN* | | | | | | | | | |
| GRAPE | 35.14 | 42.45 | 38.08 | 39.87 | 18.81 | 60.73 | 48.43 | 53.73 | 50.85 |
| IGRM | 35.23 | 41.31 | 36.45 | 40.09 | 18.97 | 61.35 | — | 59.86 | — |
| *Generative models* | | | | | | | | | |
| MIWAE | 76.27 | 72.30 | 57.30 | 77.31 | 62.64 | 57.99 | 51.10 | 46.74 | 63.26 |
| GAIN | 69.68 | 67.50 | 73.70 | 66.11 | 53.46 | 92.81 | 60.04 | 46.69 | 57.49 |
| MCFlow | 60.00 | 68.23 | 65.59 | 55.27 | 30.40 | 84.36 | 67.87 | 72.99 | 56.36 |
| TabCSDI | 83.17 | 77.42 | 54.64 | 76.93 | 75.34 | 57.11 | 46.48 | 58.53 | 66.41 |
| MissDiff | 119.49 | 56.69 | 229.19 | 55.82 | 45.62 | 55.01 | 50.62 | 48.19 | 54.41 |
| *Other* | | | | | | | | | |
| KNN | 49.25 | 49.41 | 45.52 | 49.70 | 30.81 | 50.83 | 35.15 | 46.81 | 48.53 |
| ReMasker | 34.26 | 33.93 | 36.52 | 50.21 | 16.85 | 52.60 | 40.58 | 38.24 | 30.37 |
| HyperImpute | 36.22 | 39.74 | 36.44 | 42.23 | 16.93 | 49.91 | 29.66 | 36.93 | 26.85 |
| **DIFFPUTER** | 32.66 | 32.22 | 30.39 | 29.66 | 15.92 | 49.51 | 26.61 | 35.19 | 28.22 |

Table 8: MAR, In-sample imputation RMSE.

| Method | California | Letter | Gesture | Magic | Bean | Adult | Default | Shoppers | News |
|---|---|---|---|---|---|---|---|---|---|
| *Traditional iterative* | | | | | | | | | |
| EM | 60.28 | 77.77 | 67.68 | 74.17 | 39.37 | 88.74 | 79.48 | 65.76 | 82.32 |
| MICE | 85.13 | 106.80 | 92.85 | 102.74 | 49.96 | 126.50 | 109.18 | 89.10 | 105.35 |
| MIRACLE | 105.54 | 96.64 | 104.10 | 105.48 | 100.19 | 102.18 | 107.70 | 105.52 | 105.53 |
| SoftImpute | 83.23 | 88.33 | 99.13 | 87.11 | 56.32 | 95.44 | 100.26 | 94.04 | 90.61 |
| MissForest | 72.08 | 84.20 | 75.70 | 74.75 | 51.24 | 106.97 | 83.22 | 70.15 | 93.03 |
| *Dist. match* | | | | | | | | | |
| MOT | 74.86 | 69.71 | 86.57 | 76.53 | 57.92 | 85.96 | 88.74 | 81.01 | 92.78 |
| TDM | 67.88 | 64.66 | 80.44 | 80.77 | 62.96 | 86.98 | 91.34 | 95.22 | 95.02 |
| *GNN* | | | | | | | | | |
| GRAPE | 55.73 | 57.71 | 72.88 | 66.88 | 42.33 | 86.67 | 96.19 | 80.85 | 95.50 |
| IGRM | 56.13 | 56.44 | 69.51 | 67.45 | 41.61 | 85.55 | 97.30 | 0.00 | 0.00 |
| *Generative models* | | | | | | | | | |
| MIWAE | 105.54 | 96.64 | 104.10 | 105.48 | 100.19 | 102.18 | 107.70 | 105.52 | 105.53 |
| GAIN | 98.01 | 88.89 | 112.56 | 93.54 | 79.02 | 123.26 | 103.13 | 90.26 | 117.09 |
| MCFlow | 82.84 | 90.07 | 109.49 | 89.02 | 52.83 | 112.48 | 117.18 | 86.55 | 112.21 |
| TabCSDI | 111.64 | 100.27 | 101.72 | 101.65 | 101.15 | 87.37 | 101.74 | 100.07 | 104.73 |
| MissDiff | 256.09 | 79.60 | 1183.01 | 90.38 | 97.26 | 98.17 | 121.62 | 676.10 | 1464.76 |
| *Other* | | | | | | | | | |
| KNN | 80.05 | 68.59 | 92.64 | 75.52 | 62.92 | 90.24 | 85.50 | 74.04 | 89.12 |
| ReMasker | 56.08 | 47.97 | 70.55 | 77.09 | 36.87 | 78.08 | 78.43 | 58.94 | 93.74 |
| HyperImpute | 61.94 | 57.50 | 71.98 | 69.71 | 39.01 | 88.73 | 79.84 | 58.96 | 71.46 |
| DIFFPUTER | 57.12 | 48.12 | 68.59 | 67.29 | 35.19 | 79.42 | 77.52 | 60.11 | 65.53 |

Table 9: MNAR, In-sample imputation MAE.

| Method | California | Letter | Gesture | Magic | Bean | Adult | Default | Shoppers | News |
|---|---|---|---|---|---|---|---|---|---|
| *Traditional iterative* | | | | | | | | | |
| EM | 38.71 | 56.10 | 37.49 | 50.75 | 10.48 | 56.17 | 30.06 | 42.57 | 38.93 |
| MICE | 57.63 | 81.26 | 66.33 | 77.06 | 17.12 | 98.34 | 59.55 | 76.30 | 62.27 |
| MIRACLE | 42.42 | 71.68 | 71.83 | 45.59 | 15.27 | 59.32 | 37.41 | 78.56 | 41.05 |
| SoftImpute | 64.13 | 65.58 | 55.31 | 60.76 | 28.86 | 61.11 | 43.39 | 60.12 | 63.48 |
| MissForest | 43.11 | 58.61 | 39.74 | 51.15 | 25.37 | 57.40 | 34.49 | 42.58 | 39.57 |
| *Dist. match* | | | | | | | | | |
| MOT | 42.48 | 47.70 | 44.78 | 48.05 | 25.21 | 49.39 | 35.84 | 42.20 | 48.14 |
| TDM | 35.93 | 45.22 | 43.43 | 46.24 | 20.91 | 50.29 | 41.07 | 43.48 | 53.17 |
| *GNN* | | | | | | | | | |
| GRAPE | 35.82 | 41.00 | 40.46 | 40.68 | 13.24 | 57.97 | 44.70 | 55.06 | 49.36 |
| IGRM | 34.89 | 40.18 | 40.82 | 40.92 | 13.57 | 54.52 | 0.00 | 58.14 | 0.00 |
| *Generative models* | | | | | | | | | |
| MIWAE | 75.91 | 83.18 | 60.93 | 79.34 | 62.09 | 59.03 | 56.06 | 49.34 | 64.40 |
| GAIN | 79.60 | 67.04 | 83.73 | 64.75 | 39.68 | 123.87 | 85.44 | 55.89 | 57.33 |
| MCFlow | 59.23 | 65.53 | 67.03 | 60.37 | 26.30 | 81.16 | 67.03 | 77.87 | 54.96 |
| TabCSDI | 72.26 | 77.71 | 58.16 | 75.46 | 75.69 | 62.09 | 58.88 | 60.29 | 65.88 |
| MissDiff | 52.37 | 49.11 | 325.12 | 113.14 | 38.52 | 98.34 | 63.11 | 168.17 | 75.23 |
| *Other* | | | | | | | | | |
| KNN | 54.23 | 53.21 | 48.79 | 49.26 | 24.00 | 62.99 | 40.17 | 46.27 | 47.83 |
| ReMasker | 33.22 | 33.65 | 38.98 | 75.05 | 12.94 | 47.66 | 39.44 | 39.89 | 29.24 |
| HyperImpute | 33.96 | 40.20 | 38.30 | 42.88 | 11.96 | 50.19 | 28.36 | 39.80 | 31.74 |
| DIFFPUTER | 34.08 | 33.13 | 31.54 | 40.82 | 11.42 | 48.59 | 26.94 | 37.25 | 28.51 |

Table 10: MNAR, In-sample imputation RMSE.

| Method | California | Letter | Gesture | Magic | Bean | Adult | Default | Shoppers | News |
|---|---|---|---|---|---|---|---|---|---|
| *Traditional iterative* | | | | | | | | | |
| EM | 59.75 | 75.78 | 74.88 | 76.01 | 28.90 | 92.30 | 83.26 | 66.16 | 71.03 |
| MICE | 84.36 | 105.91 | 98.00 | 106.59 | 41.91 | 132.01 | 112.55 | 91.16 | 96.87 |
| MIRACLE | 73.98 | 109.85 | 125.48 | 74.93 | 50.36 | 106.10 | 150.95 | 82.04 | 100.59 |
| SoftImpute | 102.74 | 87.02 | 103.04 | 90.21 | 48.27 | 96.65 | 99.56 | 95.59 | 83.84 |
| MissForest | 65.80 | 80.53 | 81.17 | 76.66 | 42.72 | 99.58 | 91.87 | 70.15 | 85.34 |
| *Dist. match* | | | | | | | | | |
| MOT | 70.10 | 66.95 | 88.63 | 76.35 | 49.43 | 91.16 | 93.92 | 83.08 | 86.44 |
| TDM | 59.81 | 63.17 | 86.36 | 78.16 | 47.59 | 91.81 | 96.36 | 88.88 | 94.51 |
| *GNN* | | | | | | | | | |
| GRAPE | 55.78 | 56.43 | 78.33 | 63.91 | 32.66 | 94.56 | 99.99 | 81.17 | 84.97 |
| IGRM | 55.34 | 55.32 | 78.74 | 64.30 | 32.57 | 93.64 | 97.58 | — | — |
| *Generative models* | | | | | | | | | |
| MIWAE | 102.77 | 107.54 | 111.03 | 107.10 | 96.50 | 102.70 | 113.33 | 107.99 | 114.09 |
| GAIN | 106.38 | 88.99 | 128.97 | 89.24 | 58.92 | 184.39 | 116.56 | 93.92 | 145.92 |
| MCFlow | 82.16 | 86.87 | 114.77 | 91.89 | 47.98 | 117.01 | 123.68 | 88.87 | 105.53 |
| TabCSDI | 93.59 | 101.11 | 107.65 | 100.20 | 101.99 | 98.54 | 106.69 | 102.92 | 107.07 |
| MissDiff | 137.09 | 68.98 | 6116.33 | 227.21 | 61.13 | 229.50 | 893.48 | 449.46 | 829.78 |
| *Other* | | | | | | | | | |
| KNN | 83.15 | 73.01 | 101.66 | 75.44 | 53.33 | 103.03 | 86.35 | 85.52 | 98.56 |
| ReMasker | 54.34 | 48.02 | 75.66 | 102.23 | 32.15 | 85.60 | 82.65 | 58.61 | 80.17 |
| HyperImpute | 58.93 | 57.84 | 75.84 | 69.87 | 31.24 | 89.04 | 86.96 | 61.48 | 76.42 |
| DIFFPUTER | 57.48 | 48.54 | 69.54 | 67.66 | 32.59 | 85.26 | 78.82 | 59.83 | 66.74 |

