# OpenReview forum: "DiffPuter: Empowering Diffusion Models for Missing Data Imputation"
_ICLR.cc/2025/Conference — ICLR 2025 Spotlight_

### Official Review · Reviewer_rwwt · 2024-10-18

**Soundness:** 3
**Presentation:** 3
**Contribution:** 3
**Rating:** 8
**Confidence:** 3

**Summary:**

The authors combine diffusion processes and Expectation-Maximization (EM) algorithms to propose a novel way to impute missing data when both training and tests sets contain missing data. The proposed solution is shown to target the correct conditional distribution (distribution of missing data conditional on observed ones). Imputation values are computed by taken the expectation with respect to this distribution, which is approximated by the sample mean. Experiments on ten real-world data sets show the benefit of the proposed method, compared to various state-of-the-art imputation algorithms (machine and deep learning algorithms).

**Strengths:**

The authors propose a new method to impute missing data for continuous and discrete inputs. The proposed method appears to be new, with excellent performances. An extensive literature review has been done to present and explain the previous approaches to deal with missing values via imputation. The method is clearly explained, the paper well-written, and the experiments show the benefit of the proposed method.

**Weaknesses:**

I only have three remarks:
- RMSE and MAE are measures that encourage the imputation to target the (conditional) mean or median. In both cases, the target is not a distribution but a single quantity. Recent works (https://arxiv.org/abs/2002.03860) have shown that such measures do not properly evaluate the correctness of an imputation method. Imputation score (https://arxiv.org/pdf/2106.03742) can be used instead to assess the quality of imputations. As the proposed method generates a distribution and not a single point estimate, it is likely that its performance will be higher with respect to this metric, showing that it is able to recover the underlying distribution of the data. Presenting imputation scores in the tables would definitely improve the strength of the paper, in my opinion.
- The computational performances of DiffPuter should be discussed in the main text. Table 4 is interesting, as it shows that the training time is larger, but not too important. However, the two considered data sets have few features. It would be appealing to consider larger data sets with (i) more observations and/or (ii) more variables to see how the predictive performances and the training time behave.
- I have trouble understanding the proof of Theorem 1. Notations are confused to me. Adding a table of notations, with exact definitions at the beginning of the Appendix would help. Besides, many approximations are done in the proof : l.730, 731, 750, 753. This results in the theorem being imprecise. For example, nothing is assumed about the quality of the neural network $\varepsilon_{\theta}$. What type of convergence is required for Theorem 1 to be valid? Similarly, in Theorem 1, $\sigma(T)$ is not assumed to be large, whereas it is required in the proof. Please clarify the different assumptions and the proof.

**Questions:**

- l.197: could you specify the choice of $\sigma(t)$?
- l.225-227: the paragraph does not correspond to the equation: the negative log likelihood is upper bounded by the loss plus a constant, which does not imply that optimizing the first leads to optimizing the second.
- Section 5.1, how does the method behave when different masks are present in the training and test set? Does it degrade the performances?
- Section 5.1, how were the hyperparameter chosen for the different baselines? Are these baselines comparable (in terms of number of parameters for example) with the proposed method? Could you add such a discussion in the Appendix? Could you also describe in details the missing data mechanisms used for the different settings (MAR and MNAR encapsulate a lot of data generating processes)?
- l.366-367, Can you explain the good performances of the proposed method compared to MissDiff and TabCSDI?


l.399 : "Imputaing"
l. 468 : "Gestrue"

---

> ### Author Response · Authors · 2024-11-19
> **Response to Reviewer rwwt (W1)**
>
> We thank the reviewer for the time spent reviewing our paper and for providing constructive questions and suggestions. The following is our detailed response to every question.
>
> ### W1: A novel metric, the imputation scores
> > RMSE and MAE are measures that encourage the imputation to target the (conditional) mean or median. In both cases, the target is not a distribution but a single quantity. Recent works (https://arxiv.org/abs/2002.03860) have shown that such measures do not properly evaluate the correctness of an imputation method. Imputation score (https://arxiv.org/pdf/2106.03742) can be used instead to assess the quality of imputations. As the proposed method generates a distribution and not a single point estimate, it is likely that its performance will be higher with respect to this metric, showing that it is able to recover the underlying distribution of the data. Presenting imputation scores in the tables would definitely improve the strength of the paper, in my opinion.
>
> We thank the reviewer for recommending the novel metric, and we agree that comparing the performance of different methods under this metric will significantly improve our paper. Therefore, we have conducted additional experiments for this metric, and will add these results in the revised paper.  Also, this metric has been integrated into our codebase.
>
> When using this metric, we find that the computation of the imputation score seems rather inefficient and coul take a extremely long time for the entire dataset. Therefore, we randomly sample 500 rows/samples from the in-sample imputation results. The hyperparameters are set as follows:
>
> - num.trees.per.proj to be 5
> - The minimum node size in each tree is 10 (the default for a probability RF).
> - We chose the number of projections (num.proj) adaptively to the dimension $d$ of the data set: for $d \le 6$ we used 50, for $7 ≤ d ≤ 14$ we used $100$ and for $d ≥ 15$ we used $200$.
>
> We present the imputation scores of different methods in the following table.
>
> | Datasets | Ground-Truth | DiffPuter |  HyperImpute | ReMasker | MOT | Mean |
> | ------- | ------ | ------| ------ | ------ | ----- | ---- |
> | Adult   | 0.0 | **-0.0490** | -0.5013 | -0.9878 | -1.0723 | -11.1952  |
> | Shoppers | 0.0 | **-1.5615** | -1.5997 | -3.5933 | -2.8280 | -9.5285|
> | Beijing | 0.0 | **-0.4061** | -0.4202 | -2.3349 | -1.8364 | -14.1026 |
>
> As demonstrated in the table, in terms of the imputation score metric, our DiffPuter still significantly outperforms other methods.

---

> ### Author Response · Authors · 2024-11-19
> **Response to Reviewer rwwt (W2)**
>
> ### W2: The computational performance should be discussed in the main text.
>
> > The computational performances of DiffPuter should be discussed in the main text. Table 4 is interesting, as it shows that the training time is larger, but not too important. However, the two considered data sets have few features. It would be appealing to consider larger data sets with (i) more observations and/or (ii) more variables to see how the predictive performances and the training time behave.
>
> Thank you for the suggestion. We've moved this part to Section 5.3 in the main text. In addition, we consider a novel dataset with more rows and columns, the [Covertype](https://archive.ics.uci.edu/dataset/31/covertype) dataset. The original dataset consists of 581,012 instances and 54 features, and we subsample 200,000 records such that we can complete the experiments in the rebuttal phase. The following table compares the training time and our DiffPuter's performance advantage over representative baselines (we also present the results on Adult and California for reference):
>
> | Datasets   | MOT  | TDM | GRAPE | IGRM | Hyperimpute | Remasker | DiffPuter (Ours) |
> | ------ | --------- | ------ | --------- | ------ | --------- | ------ | --------- |
> | California | 446.47s | 216.91s | 635.7s | 1267.5s | 1276.3s | 1320.1s | 1927.2s|
> | Adult  | 396.68s | 514.56s | 2347.1s | 3865.1s | 1806.9s | 1902.4s | 2142.9s |
> |Avg. Perf. advantage |  21.47\% | 21.37\% | 25.94\% | 20.97\% | 8.44\% | 10.65\% | - |
>
>
> | Datasets   | MOT  | TDM | GRAPE | IGRM | Hyperimpute | Remasker | DiffPuter (Ours) |
> | ------ | --------- | ------ | --------- | ------ | --------- | ------ | --------- |
> | CoverType | 182.5min | 235.4min | OOM | OOM | 833.6min | 873.6min | 1007.0min|
> | Perf. advantage |  24.8\%  | 23.9\% | - | - | 15.4\% | 18.7\% | - |
>
> On such a large dataset, graph-based methods like GRAPE and IGRM both face OOM (Out Of Memory) problems, while other methods that can perform mini-batch training can run normally.
>
> From the perspective of training speed, the training time required by different methods generally shows a linear relationship with the number of rows and columns in the dataset. The training time required by our DiffPuter is basically at the same level as HyperImpute and Remasker. From the perspective of imputation performance, our DiffPuter shows even greater advantages compared to other methods, which may be attributed to the huge capacity and exceptional ability of Diffusion models in modeling large-scale and high-dimensional data distributions.
>
> We will include the CoverType dataset as a benchmark dataset in our regular experiments, and add all its results to the updated paper once they are obtained.

---

> ### Author Response · Authors · 2024-11-19
> **Response to Reviewer rwwt (W3)**
>
> ### W3: Understanding of the proof of Theorem 1
>
> > I have trouble understanding the proof of Theorem 1. Notations are confused to me. Adding a table of notations, with exact definitions at the beginning of the Appendix would help. Besides, many approximations are done in the proof : l.730, 731, 750, 753. This results in the theorem being imprecise. For example, nothing is assumed about the quality of the neural network. What type of convergence is required for Theorem 1 to be valid? Similarly, in Theorem 1, $\sigma(T)$ is not assumed to be large, whereas it is required in the proof. Please clarify the different assumptions and the proof.
>
> Thank you for your criticism and suggestions. In the updated paper, we have added a new section at the beginning of the Appendix (Appendix A) to introduce all symbols and notations used in this paper.
>
> The proof of Theorem 1 in the Appendix indeed needs the help of some mild assumptions for approximation. We list these assumptions/approximation as follows:
>
> - line 730: Small reverse process time intervals: $\Delta t \rightarrow 0$, such that the difference between $x_t^{\rm mis}$ and $x_{t-\Delta t}^{\rm mis}$ is negligible.
> - line 731 should be '=' rather than '$\approx$' since Monte Carlo estimation is unbiased.
> - line 750: We require a large maximum timestep $\sigma(T) \gg 1$, such that $p(\mathbf{x}_T | \mathbf{x}_0)\approx p(\mathbf{x}_T)$
>
> The necessary assumptions have been added to the description of Theorem 1 in the revised paper. Thanks again for your criticism.
>
> Note: Line 751, Theorem 1 does not need to assume the quality of the neural network $\epsilon_{\theta}$, because Theorem 1/Lemma 1 fundamentally explains that for any denoising network $\epsilon_{\theta}$, our method can sample/generate from its induced time-dependent distribution at arbitrary time step $t$, $p_{\theta}(\mathbf{x}_t | \hat{\mathbf{x}}^{obs})$.
>
>  The statement '''the score function
> $$\nabla_{\mathbf{x}_t}\log p (\mathbf{x}_t)$$
>
> is replaced with $\varepsilon_{\theta}(\mathbf{x}_t, t)$''',
>
>  does not indicate approximation, but rather indicates that the distribution obtained by the reverse process is induced by the denoising neural network. Specifically, when $\varepsilon_{\theta}(\mathbf{x}_t, t)$ perfectly fits the score function $\nabla{\mathbf{x}_t}\log p (\mathbf{x}_t)$, the induced distribution is the ground-truth data distribution.

---

> ### Author Response · Authors · 2024-11-19
> **Response to Reviewer rwwt (Q1)**
>
> > l.197: could you specify the choice of $\sigma(t)$?
>
> In this paper, we set $\sigma(t) = t$ that is linear w.r.t. the time. We apologize for this omission and have added a supplementary explanation below Equation (2). This is a widely adopted setting for diffusion models in order to achieve better sampling efficiency (faster sampling speed)[1].
>
> > l.225-227: the paragraph does not correspond to the equation: the negative log-likelihood is upper bound by the loss plus a constant, which does not imply that optimizing the first leads to optimizing the second.
>
> Thank you for your correction. You are right; Remark 2 cannot state that optimizing the first necessarily optimizes the second. Instead, it is approximating the maximum likelihood estimation. The necessary condition is that the bound should be sufficiently tight (for example, become an equality). In fact, Theorem 2 of Song et al., 2021a [2] has extended Remark 2, showing that the equality can be achieved when score function $s_{\theta}$ perfectly aligns with the score function $\nabla_{\mathbf{x}}\log q_t(\mathbf{x})$ of a time-dependent reverse-time diffusion process with boundary distribution $q_T = \pi$ and $\mathbf{x}(0) \sim q_0$. i.e. $s_{\theta}(\mathbf{x}, t)\equiv \nabla_{\mathbf{x}}\log q_t(\mathbf{x})$.
>
> Theorem 1 also claims this condition is hard to satisfy since the score-based model $s_{\theta}(\mathbf{x}, t)$ will not exactly match $\nabla_{\mathbf{x}}\log q_t(\mathbf{x})$ everywhere (due to the capacity limitation of neural networks), therefore it is only an approximation. However, the empirical results of [2] did demonstrate that the training is actually able to improve the log-likelihood of data across multiple datasets and model architectures.
>
> We have revised the relevant paragraphs to make our claims more accurate. Thank you again for your correction.
>
> References:
>
> [[1] Karras, T., Aittala, M., Aila, T., & Laine, S. (2022). Elucidating the design space of diffusion-based generative models. Advances in neural information processing systems, 35, 26565-26577.](https://arxiv.org/abs/2206.00364)
>
> [[2] Song, Yang, et al. "Maximum likelihood training of score-based diffusion models." Advances in neural information processing systems 34 (2021): 1415-1428.](https://arxiv.org/abs/2101.09258)

---

> > ### Author Response · Authors · 2024-11-20
> > **Response to Reviewer rwwt (Q2)**
> >
> > > Section 5.1, how does the method behave when different masks are present in the training and test set? Does it degrade the performance?
> >
> > Thank you for suggesting such an interesting experimental scenario. We conduct additional experiments where in the training set we use MAR setting masks, while in the testing phase, we use MCAR setting masks. In the table below, we show the results of different methods on the Adult and Default datasets (In terms of MAE). (Due to the limited time during the rebuttal phase, we will update the results for other datasets and all other methods in later versions.)
> >
> > | Datasets | DiffPuter |  HyperImpute | ReMasker | MOT |
> > | ------- | ------ | ------| ------ | ------ |
> > | Adult   |   0.4892 | 0.5125 | 0.5391 | 0.5284 |
> > | Default | 0.2342 | 0.3195   | 0.4182 | 0.3349  |
> >
> > Compared with the results of trainig/testing on MCAR in Figure 2, the performance of all methods declines. However, our DiffPuter showes the smallest decline in performance, not exceeding one percentage point. In contrast, other methods show much larger performance drops.
> >
> > In particular, the Remasker method, as a discriminative method based on mask prediction, is significantly affected by the mask generation patterns in the original data during its learning process. The patterns learned from the training set are difficult to apply to the test set if the distributions are different. Our DiffPuter, however, focuses on learning the overall joint distribution and continuously corrects the estimation of missing values through the EM approach. Different missing patterns in the training set have minimal impact on the final learned distribution.
> >
> > > l.366-367, Can you explain the good performances of the proposed method compared to MissDiff and TabCSDI?
> >
> > Thank you for your question. We believe that the poor performance of MissDiff and TabCSDI is due to their focus on the known observed portion of the original data while ignoring the missing portion.
> >
> > - **MissDiff[1]**: MissDiff simply applies DDPM to tabular datasets containing missing data (indicated by 'NA'). To adapt to the missing data scenario, its diffusion loss (score-matching) is only applied to entries where observed data exists.
> > - **TabCSDI[2]**: TabCSDI resorts to a conditional diffusion model. It uses masks to further divide the observed data into two parts: the conditional part and the target part. It aims to learn the conditional distribution of any target part conditioned on the conditional part.
> >
> > The above two methods both separate the observed data and missing data in the original data and foucs on learning the distribution of observed data, ignoring the importance of missing data in the original data. Our method, however, utilizes the value of missing entries in the original data by treating them as latent variables and continuously updating them through the EM algorithm.
> >
> >
> > References:
> >
> > [[1] Ouyang, Y., Xie, L., Li, C., & Cheng, G. (2023). Missdiff: Training diffusion models on tabular data with missing values. arXiv preprint arXiv:2307.00467.](https://arxiv.org/pdf/2307.00467)
> >
> > [[2] Zheng, S., & Charoenphakdee, N. (2022). Diffusion models for missing value imputation in tabular data. arXiv preprint arXiv:2210.17128](https://arxiv.org/pdf/2210.17128)

---

> > > ### Comment · Reviewer_rwwt · 2024-11-25
> > >
> > > Thank you very much for your detailed answers and the work you put in the paper and the rebuttal!

---

> ### Author Response · Authors · 2024-11-19
> **Response to Reviewer rwwt (Q3-1)**
>
> ### Q3: The selection of hyperparameters for different baselines.
> > Section 5.1, how were the hyperparameter chosen for the different baselines? Are these baselines comparable (in terms of number of parameters for example) with the proposed method? Could you add such a discussion in the Appendix?
>
>
> Most of the deep learning baselines recommend the use of one set of hyperparameters for all datasets. For these methods, we directly follow their guidelines and use the default hyperparameters. For Remasker, GRAPE, and IGRM, where the model widths can be enlarged or reduced, we tried to align their model sizes but observed little impact on their performance. Below is the detailed introduction of how the hyperparameters of each baseline method is selected (and we've added a discussion part in Appendix D.6):
>
> * **ReMasker**: we use the recommended hyperparameters provided in Appendix A.2 in the original paper [1]. This set of hyperparameters is searched by tuning on the Letter dataset and is deployed for all the datasets in the original paper; hence, we follow this setting.
> * **HyperImpute**: since HyperImpute works by searching over the space of classifiers/regressors and their hyperparameters, it does not have hyperparameters itself except parameters related to the AutoML search budget. We adopt the default budget parameters of HyperImpute's official implementation for all datasets. The default budget parameters and AutoML search space are provided in https://github.com/vanderschaarlab/hyperimpute/blob/main/src/hyperimpute/plugins/imputers/plugin_hyperimpute.py and Table 5 in the original paper [2].
> *  **MOT and TDM**: There is a main hyperparameter representing the number of subset pairs sampled from the dataset for computing optimal transport loss. Sinkhorn algorithm and TDM are controlled by hyperparameter *n_iter*. While the default value is 3000 (https://github.com/BorisMuzellec/MissingDataOT/blob/6417eacfde1c1052a63568350dfec2d0373ac056/experiment.py#L42), we set it as 12000 for all datasets to ensure the algorithm converges sufficiently. For the round-robin version of the algorithm, the number of sampled pairs is controlled by *max_iter* and *rr_iter*; we adopt the default value 15, which is enough for the algorithm to converge. For the remaining hyperparameters related to network architectures, we use the default ones for all datasets.
> * **kNN**: we follow the common practice of selecting the number of nearest neighbors as $\sqrt{n}$, where $n$ is the number of samples in the dataset.
> * **GRAPE and IGRM**: we adopt the recommended set of hyperparameters used in the original paper for all datasets. For a detailed explanation of the meaning of the parameters, please see https://github.com/maxiaoba/GRAPE/blob/0ea0c59272a977d0184a8fd95178f68211455ef5/train_mdi.py#L18 for GRAPE and https://github.com/G-AILab/IGRM/blob/5cfc955daa5d0f4bbdcbad1552cfd7493dfd5fd0/main.py#L17 for IGRM.
> * **MissDiff**: since the original implementation is not available, and it is based on diffusion model, for fair comparison, we simply use the same set of hyperparameters with our DiffPuter.
> * **TabCSDI**: We follow the guide for selecting hyperparameters in the original paper (Appendix B in [3]). Specifically, we use a large version of the TabCSDI model with a number of layers set to 4 (see more detailed hyperparameters about the large TabCSDI model at https://github.com/pfnet-research/TabCSDI/blob/main/config/census_onehot_analog.yaml). For batch size, we take the official choice of batch size (8) for the breast dataset (~700 samples) as a base, and scale the batch size accordingly with the sample size of our datasets: since most of the datasets we used have the number of samples between 20000 to 40000, we scale the batch size to 256 and use it for all datasets.
> * **MCFlow**: we adopt the recommended hyperparameters provided in the official implementation for all datasets (https://github.com/trevor-richardson/MCFlow/blob/70fe137db79255bfbec07c5605ccd3fe0c52c789/main.py#L67).
>
> For the remaining classical machine learning methods, including EM, GAIN， MICE，Miracle，MissForest, and Softimpute where hyperparameters might be important. Since we use the implementations from the 'hyperimpute' package, we tune the hyperparameters within the hyperparameter space provided in the package (e.g., https://github.com/vanderschaarlab/hyperimpute/blob/e9506c7b3a1f5089f00797534e0460fd28f9730c/src/hyperimpute/plugins/imputers/plugin_missforest.py#L73). To be specific, we set the maximum budge as 50, then we sample 50 different hyperparameter combinations according to the hyperparameter space. Finally, we report the optimal performance over the 50 trials.

---

> ### Author Response · Authors · 2024-11-19
> **Response to Reviewer rwwt (Q3-2)**
>
> ## How the missing data mechanism is implemented?
>
> > Could you also describe in details the missing data mechanisms used for the different settings (MAR and MNAR encapsulate a lot of data generating processes)
> >
> We follow the methods proposed in TDM[4] to implement MAR and MNAR.
>
> - "For MAR, we first sample a subset of features (columns in $X$) that will not contain missing values and then we use a logistic model with these non-missing columns as input to determine the missing values of the remaining columns and we employ line search of the bias term to get the desired proportion of missing values."
>
> - For MNAR, we use the first approach proposed in [4], "Using a logistic model with the input masked by MCAR".  Specifically, similar to the MAR setting, we first divide the data columns into two groups, with one group serving as input for the logistic model, outputting the missing probability for the other set of columns. The difference is that after determining the missing probability of the second set, we apply MCAR to the input columns (the first set). Hence, missing values from the second set will depend on the masked values of the first set.
>
> We have also updated this content in Section D.3 of the revised paper's Appendix.
>
> References:
>
> [[1] Tianyu Du, Luca Melis, and Ting Wang. Remasker: Imputing tabular data with masked autoencoding. In International Conference on Learning Representations, 2024.](https://arxiv.org/abs/2309.13793)
>
> [[2] Daniel Jarrett, Bogdan C Cebere, Tennison Liu, Alicia Curth, and Mihaela van der Schaar. Hyperimpute: Generalized iterative imputation with automatic model selection. In International Conference on Machine Learning, pp. 9916–9937. PMLR, 2022.](https://arxiv.org/abs/2206.07769)
>
> [[3] Shuhan Zheng and Nontawat Charoenphakdee. Diffusion models for missing value imputation in tabular data. In NeurIPS 2022 First Table Representation Workshop, 2022.](https://arxiv.org/abs/2210.17128)
>
> [[4] Zhao, H., Sun, K., Dezfouli, A., & Bonilla, E. V. (2023, July). Transformed distribution matching for missing value imputation. In International Conference on Machine Learning (pp. 42159-42186). PMLR.](https://arxiv.org/abs/2302.10363)

---

### Official Review · Reviewer_TxpG · 2024-10-28

**Soundness:** 2
**Presentation:** 3
**Contribution:** 2
**Rating:** 8
**Confidence:** 4

**Summary:**

This paper introduces an adaptation of the EM algorithm for missing data imputation, leveraging advanced diffusion-based models to perform precise density estimation in the M-step and provide robust imputations in the E-step, inspired by the RePaint algorithm. The authors reference theoretical analyses from prior work to support the use of diffusion models for density estimation and prove a theorem demonstrating that E-step samples can be drawn from the true conditional distribution. Extensive empirical evaluations highlight the proposed method’s robustness and superiority over various baseline approaches, many of which do not incorporate the EM algorithm.

**Strengths:**

- Robust imputation method based on EM.
- Well written and structured.
- The method is theoretically grounded.
- The empirical analysis is extensive.

**Weaknesses:**

- Motivation appears to overlook recent work.
- Experimental section lacks fair comparison and clarity.
- Discussion of limitations is lacking.
- Given these weaknesses, the contribution is not strongly justified.


------- Post rebuttal update -------

All the weaknesses were thoroughly addressed in the rebuttal provided by the authors. I appreciate their efforts and the detailed responses, which resolved all my concerns.

**Questions:**

### Motivation appears to overlook recent work

- The two main issues presented as motivation for this work are unclear. The paper claims that generative imputation methods (i) require joint estimation of observed and missing data distributions and (ii) struggle with conditional inference. I find both statements questionable. Numerous studies adapt deep generative models to estimate only the observed data distribution [1-5], which could serve in the M-step of an EM algorithm. Some of these are even referenced in this paper. Moreover, all of these methods allow for straightforward Monte Carlo estimation of $\mathbb{E}[p(\mathbf{x}_m | \mathbf{x}_o)]$ for the E-step, similar to the proposed diffusion-based model. For instance, a more robust importance-weighted estimator is proposed in [4] (see Eq. (12)).

- This brings me to a second point: if multiple DGMs could, indeed, replace diffusion models within the EM framework, how is diffusion specifically justified for tabular data? This approach might be advantageous for high-dimensional data, where diffusion models effectively approximate $p(\mathbf{x})$ and avoid lossy compression (as in VAEs). However, given the lower-dimensional datasets studied here, it remains unclear why a VAE-based approach, for example, wouldn’t perform as well as diffusion.

### Experimental section lacks fair comparison and clarity

- Based on my earlier points, I would expect an ablation study comparing different DGMs within the EM algorithm. The baselines in the current experiments appear to rely on simple placeholder values for missing data (e.g., zero or mean imputation), effectively completing only one M-step. This is likely to produce suboptimal results, so a performance gap seems unsurprising.

- The assertion "Traditional machine learning methods are still powerful imputers" would benefit from supporting references. I am skeptical, as optimal validation could be harder to achieve in probabilistic settings.

- The claim that generative methods excel on continuous data requires clarification. Here, the diffusion model seems to assume Gaussianity across all dimensions, using $argmax$ as a proxy to obtain discrete outputs, which is not the optimal to model heterogeneous data [2, 3, 5].

- The statement "imputation methods are specifically designed for in-sample imputation and cannot be applied to the out-of-sample setting" also needs elaboration. As mentioned, many DGMs designed for missing data can perform out-of-sample imputation.

- In Figure 2, MissDiff appears to fail or encounter out-of-memory issues. This is surprising, as MissDiff’s architecture is similar to the diffusion network used here.

### Discussion of limitations is lacking

- The main text does not discuss limitations, particularly the high computational cost. A brief note is found in the Appendix, but this isn’t referenced within the primary text. DiffPuter’s approach requires retraining the diffusion model $k$ times, so application to high-dimensional data (e.g., images) would be computationally intense relative to alternatives. I am also curious if the M-step converges faster with higher values of $k$, as this could enhance efficiency.

### Other minor questions

- Figure 6: Why does error decrease as observed data ratio drops? I found the final paragraph of Section 5 somewhat unclear; further clarification here would be helpful.

### Typos
- Line 515: Change *", Reducing"* to *", reducing"*.


### References

[1] Ma, Chao, et al. "EDDI: Efficient Dynamic Discovery of High-Value Information with Partial VAE." International Conference on Machine Learning. PMLR, 2019.

[2] Ma, Chao, et al. "VAEM: a deep generative model for heterogeneous mixed type data." Advances in Neural Information Processing Systems 33 (2020): 11237-11247.

[3] Peis, Ignacio, Chao Ma, and José Miguel Hernández-Lobato. "Missing data imputation and acquisition with deep hierarchical models and hamiltonian monte carlo." Advances in Neural Information Processing Systems 35 (2022): 35839-35851.

[4] Mattei, Pierre-Alexandre, and Jes Frellsen. "MIWAE: Deep generative modelling and imputation of incomplete data sets." International conference on machine learning. PMLR, 2019.

[5] Nazabal, Alfredo, et al. "Handling incomplete heterogeneous data using VAEs." Pattern Recognition 107 (2020): 107501.

---

> ### Author Response · Authors · 2024-11-20
> **Response to Reviewer TxpG (Q1)**
>
> ### Q1: Motivation of the paper
>
> > Motivation appears to overlook recent work. The two main issues presented as motivation for this work are unclear. If multiple DGMs could, indeed, replace diffusion models within the EM framework, how is diffusion specifically justified for tabular data?
>
> We apologize for the confusion caused by the lack of clarity in the motivation section. Our motivation is indeed based on "the necessity of using diffusion models as deep generative models to learn tabular distributions". This is because if a generative model cannot accurately learn the data distribution, even if it can perform accurate conditional sampling, the obtained imputation values will be inaccurate. And in the context of EM, such errors may further accumulate.
>
> Regarding the performance comparison of different types of generative models on tabular data generation tasks, it has been studied in recent works[1,2]. Through extensive experiments, the conclusion is that early deep generative models, such as GANs and VAEs[3,4,5], are quite poor at learning tabular data distributions, and their generated samples struggle to faithfully recover the ground-truth distribution (poor capacity in density estimation). In contrast, Diffusion-based models[1,2] have shown performance far exceeding that of VAE, GAN, and other methods.
>
> Although Diffusion models, as a type of deep generative model, have achieved SOTA performance in tabular data generation, they directly model the joint distribution of all columns, making it difficult to perform conditional inference. This forms our complete motivation.
>
> The motivation in the initial version didn't clearly explain this point, so we made substantial modifications to the introduction to make our motivation clearer (which has been updated in the revised paper). The key points are as follows:
>
> - **Motivation of using EM algorithm**: The biggest challenge in applying deep generative models to missing data imputation is the inability to observe the complete distribution. The EM algorithm addresses the incomplete likelihood issue by iteratively refining the values of the missing data.
>
> - **Challenges of combine EM algorithm and DGMs**
>     - EM algorithm consists of an M-step (density estimation) and E-step (conditional inference).
>     - Achieving both a good E step and M-step is a dilemma.
>     - Existing DGMs (Deep Generative Models) that can perform conditional inference (such as VAE) cannot capture tabular data distribution well. Meanwhile, SOTA methods that can faithfully recover data distribution (such as Diffusion) struggle to perform conditional inference.
>
> - **Our contribution**: Our method explores a path that makes diffusion models compatible with the EM framework for missing data imputation.
>
>
> Therefore, although other DGMs can technically replace the Diffusion model in our framework, in terms of empirical performance, the diffusion model is irreplaceable. In the following response to Q2, we have conducted supplementary experiments to demonstrate the importance of the Diffusion model in our framework.
>
> References:
>
> [[1] Kotelnikov, A., Baranchuk, D., Rubachev, I., & Babenko, A. (2023, July). Tabddpm: Modelling tabular data with diffusion models. In International Conference on Machine Learning (pp. 17564-17579). PMLR.](https://arxiv.org/abs/2209.15421)
>
> [[2] Zhang, H., Zhang, J., Shen, Z., Srinivasan, B., Qin, X., Faloutsos, C., ... & Karypis, G. Mixed-Type Tabular Data Synthesis with Score-based Diffusion in Latent Space. In The Twelfth International Conference on Learning Representations.](https://arxiv.org/abs/2310.09656)
>
> [[3] Xu, L., Skoularidou, M., Cuesta-Infante, A., & Veeramachaneni, K. (2019). Modeling tabular data using conditional gan. Advances in neural information processing systems, 32.](https://proceedings.neurips.cc/paper/2019/hash/254ed7d2de3b23ab10936522dd547b78-Abstract.html)
>
> [[4] Liu, T., Qian, Z., Berrevoets, J., & van der Schaar, M. (2023). GOGGLE: Generative modelling for tabular data by learning relational structure. In The Eleventh International Conference on Learning Representations.](https://openreview.net/forum?id=fPVRcJqspu)
>
> [[5] Richardson, T. W., Wu, W., Lin, L., Xu, B., & Bernal, E. A. (2020). Mcflow: Monte carlo flow models for data imputation. In Proceedings of the IEEE/CVF conference on computer vision and pattern recognition (pp. 14205-14214).](https://openaccess.thecvf.com/content_CVPR_2020/html/Richardson_McFlow_Monte_Carlo_Flow_Models_for_Data_Imputation_CVPR_2020_paper.html)

---

> ### Author Response · Authors · 2024-11-20
> **Response to Reviewer TxpG (Q2-1)**
>
> ### Q2: The experimental section lacks fair comparison and clarity
>
> > Based on my earlier points, I would expect an ablation study comparing different DGMs within the EM algorithm. The baselines in the current experiments appear to rely on simple placeholder values for missing data (e.g., zero or mean imputation), effectively completing only one M-step. This is likely to produce suboptimal results, so a performance gap seems unsurprising.
>
> Thanks for your suggestion. We've conducted additional experiments combining EM algorithm with these DGMs. In the following table, we present the performance (MAE metric) of the proposed DiffPuter with EM+other DGMs, i.e., MIWAE[1] and HIWAE[2]. For comparion, we also present the performance of DiffPuter at different EM iteratons.
>
> | EM + MIWAE | k = 1 | k = 2 |  k = 3 | k = 4 | k = 5 | k = 6 |
> | -------  | ------ | ------| ------ | ------ | ----- | ---- |
> | Adult | 0.5763	| 0.5670 | 0.5661 | 0.5661 | 0.5661 | 0.5661 |
> | Beijing | 0.5575 | 0.5472 | 0.5452 | 0.5445 | 0.5444 | 0.5444 |
> | Default  | 0.5194 | 0.5050 | 0.5009 | 0.4997 | 0.4994 | 0.4993 |
> | News | 0.6349 | 0.6239 | 0.6197 | 0.6181 | 0.6174 | 0.6171 |
> | Shoppers | 0.5047 | 0.4713 | 0.4604 | 0.4569 | 0.4558 | 0.4554 |
>
>
> | EM + HIWAE | k = 1 | k = 2 |  k = 3 | k = 4 | k = 5 | k = 6 |
> | -------  | ------ | ------| ------ | ------ | ----- | ---- |
> | Adult | 0.6155 | 0.6167 | 0.6183 | 0.6017 | 0.5881 |0.5974 |
> | Beijing | 0.4996 | 0.5018 | 0.5015 | 0.5104 | 0.5088 | 0.5234 |
> | Default  | 0.3989 | 0.4181 | 0.4311 | 0.4039 | 0.4169 | 0.4314 |
> | News | 0.5032 |0.5022 |0.4988 |0.5111	| 0.5173 |0.5222 |
> | Shoppers |0.4707 | 0.5141 | 0.5036 | 0.4898 | 0.4913 | 0.4961|
>
> | DiffPuter | k = 1 | k = 2 |  k = 3 | k = 4 | k = 5 | k = 6 |
> | -------  | ------ | ------| ------ | ------ | ----- | ---- |
> | Adult | 0.4820 | 0.3829 | 0.3574 | 0.3499 | 0.3426 | 0.3425|
> | Beijing | 0.4126 | 0.3421 | 0.3046 | 0.2861 | 0.2792 | 0.2784 |
> | Default  |  0.3705 | 0.3115 | 0.2821 | 0.2718 | 0.2686 | 0.2661 |
> | News |  0.3945 | 0.3419 | 0.3156 | 0.2969 | 0.2876 | 0.2855 |
> | Shoppers | 0.4345 | 0.3782 | 0.3582 | 0.3559 | 0.3499 | 0.3485 |
>
> As demonstrated, even when considering only the first round of iteration, our DiffPuter's performance is far better than MIWAE and HIWAE, which demonstrates the irreplaceable ability of Diffusion models to reconstruct ground-truth data distribution compared to other DGMs. As the iteration steps increase, DiffPuter's performance shows significant improvement. MIWAE's performance also shows some improvement, but the magnitude is quite small. HIWAE's performance shows no significant improvement, and instead appears to be fluctuating. This shows that even with more EM iterations, if the data density is not correctly learned, the imputation performance remains difficult to improve.
>
> The implementation code for these two models has been added to our codebase. We will include these DGM+EM as variants of DiffPuter in the ablation study, after all the new experiments are completed.
>
> References:
>
> [[1] Mattei, Pierre-Alexandre, and Jes Frellsen. "MIWAE: Deep generative modeling and imputation of incomplete data sets." International conference on machine learning. PMLR, 2019.](https://arxiv.org/abs/1812.02633)
>
> [[2] Nazabal, Alfredo, Pablo M. Olmos, Zoubin Ghahramani, and Isabel Valera. "Handling incomplete heterogeneous data using vaes." Pattern Recognition 107 (2020): 107501.](https://www.sciencedirect.com/science/article/pii/S0031320320303046?casa_token=KfIqTtTi4z0AAAAA:UirFD3qSZIr5pkXucB6gDP5DyzQmwCSp3HocxcRkQ-Nd7Tg1d4L91GkSEEnKeV0zW0x_bah8hQ)

---

> ### Author Response · Authors · 2024-11-20
> **Response to Reviewer TxpG (Q2-2)**
>
> > The assertion "Traditional machine learning methods are still powerful imputers" would benefit from supporting references. I am skeptical, as optimal validation could be harder to achieve in probabilistic settings.
>
> The assertion "Traditional machine learning methods are still powerful imputers" is first observed from the empirical performance in our Figure 2. In Figure 2, we observe that:
>
> - Simple machine learning model, such as simple vanilla EM, can already achieve performance exceeding many deep learning methods (such as GRAPE, IGRM, MOT, TDM).
> - HyperImpute, an AutoML method that iteratively incorporates multiple machine learning-based (e.g., tree-based) imputation method ranks the second among all the baseline methods.
>
> In addition, several recent works have also highlighted the of importance and efficacy of traditional ML methods in tabular data [1,2,3,4], and we have added them in the revised version of paper.
>
> References:
>
> [[1] Jolicoeur-Martineau, Alexia, Kilian Fatras, and Tal Kachman. "Generating and imputing tabular data via diffusion and flow-based gradient-boosted trees." International Conference on Artificial Intelligence and Statistics. PMLR, 2024.](https://arxiv.org/abs/2309.09968)
>
> [[2] McCarter, Calvin. "Unmasking Trees for Tabular Data." arXiv preprint arXiv:2407.05593 (2024).](https://arxiv.org/abs/2407.05593)
>
> [[3] Lalande, Florian, and Kenji Doya. "Numerical data imputation: Choose kNN over deep learning." International Conference on Similarity Search and Applications. Cham: Springer International Publishing, 2022.](https://link.springer.com/chapter/10.1007/978-3-031-17849-8_1)
>
> [[4] Suh, Heajung, and Jongwoo Song. "A comparison of imputation methods using machine learning models." Communications for Statistical Applications and Methods 30.3 (2023): 331-341.](http://www.csam.or.kr/journal/view.html?doi=10.29220/CSAM.2023.30.3.331)
>
> > The claim that generative methods excel on continuous data requires clarification. Here, the diffusion model seems to assume Gaussianity across all dimensions, using as a proxy to obtain discrete outputs, which is not the optimal to model heterogeneous data.
>
> This is another summary observation from our empirical results. We found that our method, DiffPuter's imputation performance on continuous data, often far exceeds discriminative methods (such as Remasker, GRAPE, and MOT), but for discrete data, the lead margin is very small.
>
> Your statement is correct - the diffusion model indeed assumes Gaussian noise for each dimension. Therefore, it might only be suitable for continuous data and not for discrete data (and heterogeneous data). Our DiffPuter uses the simplest one-hot encoding to handle discrete data, and has already achieved good results. Studying different encoding methods is out of the scope of this paper, but we are interested in combining them with potentially better encoding methods if they are available.
>
>
> > The statement "imputation methods are specifically designed for in-sample imputation and cannot be applied to the out-of-sample setting" also needs elaboration. As mentioned, many DGMs designed for missing data can perform out-of-sample imputation.
>
> We apologize for writing this part too briefly. Here, we want to emphasize that several SOTA discriminative deep learning methods cannot be applied to out-of-sample imputation setting:
>
> - TDM [1] treats the values of missing entries as learnable parameters within the model, therefore it can only perform imputation on missing data in the training set and cannot be applied to the test set.
> - IGRM [2] needs to iteratively construct an implicit social network between all samples during training. For samples in the test set, they cannot be added to the existing network, therefore it doesn't work in the out-of-distribution setting.
>
> We agree that general deep generative models can naturally be applied to out-of-distribution settings. However, considering their suboptimal performance in the in-sample imputation setting (Figure 1 and Table 1), we didn't make comparisons with them. We will add the full comparison in the revised version.
>
> References:
>
> [[1] Zhao, He, et al. "Transformed distribution matching for missing value imputation." International Conference on Machine Learning. PMLR, 2023.](https://proceedings.mlr.press/v202/zhao23h.html)
>
> [[2] Zhong, Jiajun, Ning Gui, and Weiwei Ye. "Data imputation with iterative graph reconstruction." Proceedings of the AAAI Conference on Artificial Intelligence. Vol. 37. No. 9. 2023.](https://ojs.aaai.org/index.php/AAAI/article/view/26348)

---

> ### Author Response · Authors · 2024-11-20
> **Response to Reviewer TxpG (Q2-3 )**
>
> > In Figure 2, MissDiff appears to fail or encounter out-of-memory issues. This is surprising, as MissDiff’s architecture is similar to the diffusion network used here.
>
> MissDiff fails on these datasets (much larger MAE/RMSE values than other methods). In fact, MissDiff's original paper[1] has contradictory descriptions of the model: in the first half of the paper, it states that $m = 1$ indicates observed entries, while in the training section, it states that $m = 1$ indicates missing entries. Finally, the diffusion score-matching loss is only calculated on entries where $m = 1$.
>
> Since the paper does not provide implementation code, we can only reproduce the method based on the paper's textual description. Considering that calculating score-matching loss on missing entries is meaningless (since we don't know the ground-truth values at all), in our reproduction we calculate the loss on observed entries.
>
> Since MissDiff only utilizes the information from partially observed data during training while completely ignoring the missing parts in the form of masks, the data distribution it learns is inherently incomplete (i.e., concentrating only on the observed part). Thus, it's not surprising that it shows such poor imputation performance on the missing part.
>
> References:
>
> [[1] Ouyang, Y., Xie, L., Li, C., & Cheng, G. (2023). Missdiff: Training diffusion models on tabular data with missing values. arXiv preprint arXiv:2307.00467.](https://arxiv.org/pdf/2307.00467)

---

> ### Author Response · Authors · 2024-11-20
> **Response to Reviewer TxpG (Q3 & Q4)**
>
> ### Q3: Discussion of Limitations is lacking (Training time)
>
> > The main text does not discuss limitations, particularly the high computational cost. A brief note is found in the Appendix, but this isn’t referenced within the primary text. DiffPuter’s approach requires retraining the diffusion model times, so application to high-dimensional data (e.g., images) would be computationally intense relative to alternatives. I am also curious if the M-step converges faster with higher values of $k$, as this could enhance efficiency.
>
> We thank the reviewer for the suggestion, and we have moved the comparison of training time in Section 5.3 in the main text in the revised paper.
>
> The computational efficiency of this method when applied to high-dimensional data (such as images) could indeed be a problem, because training diffusion models on image data is already inefficient. And the EM algorithm will repeat training this diffusion model many times.
>
> > I am also curious if the M-step converges faster with higher values of $k$, as this could enhance efficiency.
>
>
> Thank you for raising this interesting question. In our original experiments, for each M-step, the parameters of the diffusion model are randomly reinitialized, and the model is trained from scratch, so the training time for each step is actually about the same.
>
> Your point is very reasonable. If in the new M-step, we could continue training based on the diffusion model parameters obtained from the previous M-step, the number of steps needed for model convergence could indeed be greatly reduced, thereby improving training speed.
>
> To verify this, we have conducted additional experiments using the new training strategy. In the following table, we present the number of training epochs for the convergence (loss patience of 200 = 0.2k) of diffusion model at different EM iterations.
>
> | Datasets | k = 1 | k = 2 |  k = 3 | k = 4 | k = 5 | k = 6 |
> | ------- | ------ | ------| ------ | ------ | ----- | ---- |
> | Adult   | 4.4k   |  1.2k   | 0.5k  | 0.3k  | 0.2k | 0.2k  |
> | California | 3.7k |  0.8k |  0.4k |  0.3k  | 0.3k | 0.2k  |
>
> From the table results, we can see that as EM iterations increase, the convergence speed of the diffusion model indeed becomes faster and faster, and the training time of the second iteration's diffusion model can already be reduced by over 70% compared to the first iteration. By the 5th and 6th iterations, the diffusion model has almost converged right through the beginning a few steps, and the imputation results remain almost unchanged.
>
> Regarding the overall training time, since the sampling time of the E-step remains constant, DiffPuter's overall training speed can be improved by 2 times (doubled), making it much faster than the other competitive baseline methods IGRM, Hyperimpute, and Remasker.
>
>
> ### Q4: Other minor questions
>
> > Figure 6: Why does error decrease as observed data ratio drops? I found the final paragraph of Section 5 somewhat unclear; further clarification here would be helpful.
>
> We feel sorry about causing your misunderstanding about Figure 6. Note that for the y-axis in Figure 6, higher values indicate lower MAE. Therefore, when the observed ratio decreases, MAE actually increases (indicating worse imputation performance). We've updated the corresponding paragraph to make it clearer.

---

> ### Comment · Reviewer_TxpG · 2024-11-23
> **Rebuttal Response**
>
> I thank the reviewers for their detailed feedback and the additional experiments provided during the rebuttal process. I greatly appreciate the effort and consideration that went into addressing my concerns. While many of my initial questions have been clarified, some issues still remain, mainly about the motivation of the paper:
>
> - I continue to find the justification for using diffusion models on tabular data unconvincing. Specifically, the claim that “previous generative models are poor at imputation” is unfair. There is extensive literature, not cited in the paper, demonstrating that models such as VAEs can achieve competitive results in imputation tasks on tabular data. For example, [2] and [3] build on [5] to successfully improve heterogeneous data modeling, achieving accurate imputation results. Furthermore, both TabDDPM and TABSYN, which apply a similar technique for adapting to heterogeneous data, also omits these critical references.
>
> - In your rebuttal, you state: “even if it—a generative model—can perform accurate conditional sampling, the obtained imputation values will be inaccurate.” However, it’s important to note that the deep generative models referenced above are trained on $p(x_o)$, where $x_o$ is artificially constructed by adding random missingness to the available data during training. This key trick, assumed in these works [1-5], significantly enhances imputation accuracy.
>
> As I emphasized in my initial review and again in this rebuttal, other deep generative models, such as the ones that demonstrated competitive imputation results, [2,3] could competitively replace the proposed Diffusion model for this task. I appreciate that the authors included some of the references as baselines, and I do not necessarily expect the rest to be included. However, although this comparison does not diminish the significance or novelty of DiffPutter, I simply find misleading to claim in the paper that VAEs and GANs perform poorly at imputation. This statement is inaccurate and should be revised to reflect the broader literature.

---

> ### Author Response · Authors · 2024-11-24
> **Thank you for your feedback!**
>
> We thank the reviewer for the further response. We acknowledge that we overlooked several important papers in our previous literature survey, which may have led to some inaccurate statements in the introduction section. In the latest revised paper, we have modified the relevant descriptions, added citations to these related works, and highlighted some of their advantages in missing data imputation.
>
> Regarding the importance of diffusion models in our framework empirically, we are currently conducting additional experiments replacing diffusion models with the two advanced VAE-based methods you have mentioned. Once these experiments are completed, we will add comprehensive experimental explanations in the ablation studies section. In addition, we will add one-EM-step versions of these methods as the general baselines for comparison (in Figure 2 and Table 1).
>
> Again, we thank you for your constructive comments.

---

> ### Comment · Reviewer_TxpG · 2024-11-24
> **Thanks to the authors for their thorough review**
>
> Thank you for your response. I appreciate the corrections and the effort to more comprehensively describe the related literature. While adding the suggested baselines would enhance the significance of the paper, I understand the timing constraints of the rebuttal process. In its current form, I believe the paper is acceptable.
>
> As a result, I have upgraded my score and will consider raising it further if comparisons with the aforementioned methods are included in the final version. In any case, I will recommend the paper for acceptance.

---

> > ### Author Response · Authors · 2024-11-24
> > **Response to the Reviewer**
> >
> > Thank you for your quick reply! And thank you so much for providing valuable feedback and recognizing our efforts!

---

> > ### Author Response · Authors · 2024-12-02
> > **Additonal results of combining EM with VAEM and HH-VAEM**
> >
> > We thank the reviewer for your patient waiting. We have finally implemented the combination version of the other two models (VAEM/HH-VAEM) and EM, and conducted the corresponding experiments. Their code has been updated to our anonymous code repository. Since we cannot update the PDF file now, we will add all the results of these methods to the main experiments in the next version. The table below shows the results of the Ablation study.
> >
> > | EM + MIWAE | k = 1 | k = 2 |  k = 3 | k = 4 | k = 5 | k = 6 |
> > | -------  | ------ | ------| ------ | ------ | ----- | ---- |
> > | Adult | 0.5763	| 0.5670 | 0.5661 | 0.5661 | 0.5661 | 0.5661 |
> > | Beijing | 0.5575 | 0.5472 | 0.5452 | 0.5445 | 0.5444 | 0.5444 |
> > | Default  | 0.5194 | 0.5050 | 0.5009 | 0.4997 | 0.4994 | 0.4993 |
> > | News | 0.6349 | 0.6239 | 0.6197 | 0.6181 | 0.6174 | 0.6171 |
> > | Shoppers | 0.5047 | 0.4713 | 0.4604 | 0.4569 | 0.4558 | 0.4554 |
> >
> >
> > | EM + HIWAE | k = 1 | k = 2 |  k = 3 | k = 4 | k = 5 | k = 6 |
> > | -------  | ------ | ------| ------ | ------ | ----- | ---- |
> > | Adult | 0.6155 | 0.6167 | 0.6183 | 0.6017 | 0.5881 |0.5974 |
> > | Beijing | 0.4996 | 0.5018 | 0.5015 | 0.5104 | 0.5088 | 0.5234 |
> > | Default  | 0.3989 | 0.4181 | 0.4311 | 0.4039 | 0.4169 | 0.4314 |
> > | News | 0.5032 |0.5022 |0.4988 |0.5111	| 0.5173 |0.5222 |
> > | Shoppers |0.4707 | 0.5141 | 0.5036 | 0.4898 | 0.4913 | 0.4961|
> >
> > | EM + VAEM | k = 1 | k = 2 |  k = 3 | k = 4 | k = 5 | k = 6 |
> > | -------  | ------ | ------| ------ | ------ | ----- | ---- |
> > | Adult | 0.5568 | 0.5398 | 0.5353 | 0.5530 | 0.5557 |0.5492 |
> > | Beijing | 0.4793 | 0.4489 | 0.4345 | 0.4340 | 0.4451 | 0.4440 |
> > | Default  | 0.4292 | 0.4216 | 0.4357 | 0.4039 | 0.4404 | NaN |
> > | News | 0.5204 |0.5032 |0.5068 |0.4971	| 0.4976 |0.5045 |
> > | Shoppers |0.4626 | 0.4414 | 0.4359 | 0.4304 | 0.4537 | 0.4362 |
> >
> >
> > | EM + HH-VAEM | k = 1 | k = 2 |  k = 3 | k = 4 | k = 5 | k = 6 |
> > | -------  | ------ | ------| ------ | ------ | ----- | ---- |
> > | Adult | 0.5673 | 0.5644 | 0.5520 | 0.5529 | 0.5500 |0.5402 |
> > | Beijing | 0.5025 | 0.4978 | 0.4839 | 0.5093 | 0.4867 | 0.4821 |
> > | Default  | NaN | NaN | NaN | NaN | NaN | NaN |
> > | News | NaN | NaN | NaN | NaN	| NaN | NaN |
> > | Shoppers |0.4589 | 0.4225 | 0.4127 | 0.4240 | 0.4277 | 0.4262 |
> >
> >
> >
> > | DiffPuter | k = 1 | k = 2 |  k = 3 | k = 4 | k = 5 | k = 6 |
> > | -------  | ------ | ------| ------ | ------ | ----- | ---- |
> > | Adult | 0.4820 | 0.3829 | 0.3574 | 0.3499 | 0.3426 | 0.3425|
> > | Beijing | 0.4126 | 0.3421 | 0.3046 | 0.2861 | 0.2792 | 0.2784 |
> > | Default  |  0.3705 | 0.3115 | 0.2821 | 0.2718 | 0.2686 | 0.2661 |
> > | News |  0.3945 | 0.3419 | 0.3156 | 0.2969 | 0.2876 | 0.2855 |
> > | Shoppers | 0.4345 | 0.3782 | 0.3582 | 0.3559 | 0.3499 | 0.3485 |
> >
> > We encountered some difficulties when implementing EM + HH-VAEM, mainly because when the batch size was large, we observed that the training loss would often become NaN. We could only try using a smaller batch size, but this issue remains unresolved in the News and Default datasets.
> >
> > Looking at the experimental results, VAEM / HH-VAEM generally achieved better results than MIWAE and HIWAE, both in terms of their base model (single-iteration) and multiple-iteration EM. However, their performance is still inferior to the proposed DiffPuter.

---

### Official Review · Reviewer_8jb1 · 2024-11-04

**Soundness:** 3
**Presentation:** 3
**Contribution:** 3
**Rating:** 8
**Confidence:** 3

**Summary:**

The missing value imputation is a very important problem in both machine learning and statistics. Although deep generative imputation methods have shown promise, there is still substantial room for improvement, even for matching the performance of some traditional machine learning approaches. This paper introduces DIFFPUTER, a tailored diffusion model combined with the Expectation-Maximization (EM) algorithm for missing data imputation, and shows its promising performance on a variety of datasets,

**Strengths:**

1. Theoretical analysis: DIFFPUTER’s training step corresponds to the maximum likelihood estimation of data density (M-step), and its sampling step represents the Expected A Posteriori estimation of missing values (E-step).
2. Extensive experiments that demonstrate the good performance, as compared with existing baselines, of the proposed method across various datasets.

**Weaknesses:**

the computational complexity is not explicitely discussed or compared on the numerical experiments, see details below.

**Questions:**

This paper is well-written and easy to understand. This work combines EM with the diffusion model to improve the potential inconsistency caused by missing values in the training process of diffusion models. Since EM is used with K iterations and N number of samples in the E-step, it would be beneficial to also compare the computational complexity (time complexity) of the proposed method with other diffusion-type methods, either in the discussion of the number of operations or comparing the running time in some of the numerical experiments. This would offer more insights into the proposed methods' performance from different perspectives.

---

> ### Author Response · Authors · 2024-11-21
> **Response to Reviewer 8jb1**
>
> We greatly appreciate the reviewer for taking the time to provide such constructive comments and suggestions. Below are our detailed responses to the questions you raised.
>
> ### Comparison of complexity
>
> > Since EM is used with K iterations and N number of samples in the E-step, it would be beneficial to also compare the computational complexity (time complexity) of the proposed method with other diffusion-type methods, either in the discussion of the number of operations or comparing the running time in some of the numerical experiments. This would offer more insights into the proposed methods' performance from different perspectives.
>
> Thanks for your suggestion. Here we analyze the computational complexity of our DiffPuter and two other Diffusion models, TabCSDI[1] and MissDiff[2].
>
> In the training of diffusion model, our DiffPuter and TabCSDI, MissDiff have the same level of time complexity, because they essentially all compute diffusion loss on the input data. The difference is that MissDiff and TabCSDI use 0/1 masks to compute loss only on certain entries, while our DiffPuter computes loss on all entries.
>
> During the sampling phase, all methods have the same complexity, because both TabCSDI and MissDiff need to sample many samples and take their mean as the final imputation value.
>
> Considering that our method needs to repeat this process K times, the overall computational complexity of our method is approximately K times that of MissDiff and TabCSDI.
>
> The actual training time is influenced by many factors. For example, since our DiffPuter uses a lightweight MLP model as the denoising neural network, while TabCSDI uses Transformers, in practice DiffPuter's convergence speed per iteration is much faster than TabCSDI.
>
> Considering the huge performance improvement that our DiffPuter brings compared to TabCSDI and MissDiff (approximately 50% improvement on MAE and RMSE), this increase in complexity is worthwhile.
>
> The actual training time is influenced by many factors. For example, since our DiffPuter uses a lightweight MLP model as the denoising neural network, while TabCSDI uses Transformers, in practice DiffPuter's convergence speed per iteration is much faster than TabCSDI.
>
> In Appendix D.1 (Section 5.3 in the revised version), we compared the training time of DiffPuter with other SOTA imputation methods. Overall, our method has similar training speed to other SOTA methods, but brings average performance improvements of 8% to 25%.
>
> | Datasets   | MOT  | TDM | GRAPE | IGRM | Hyperimpute | Remasker | DiffPuter (Ours) |
> | ------ | --------- | ------ | --------- | ------ | --------- | ------ | --------- |
> | California | 446.47s | 216.91s | 635.7s | 1267.5s | 1276.3s | 1320.1s | 1927.2s|
> | Adult  | 396.68s | 514.56s | 2347.1s | 3865.1s | 1806.9s | 1902.4s | 2142.9s |
> |Avg. Perf. advantage |  $21.47\%$ | $21.37\%$ | $25.94\%$ | $20.97\%$ | $8.44\%$ | $10.65\%$ | - |
>
>
> References:
>
> [[1] Zheng, S., & Charoenphakdee, N. (2022). Diffusion models for missing value imputation in tabular data. arXiv preprint arXiv:2210.17128](https://arxiv.org/pdf/2210.17128)
>
> [[2] Ouyang, Y., Xie, L., Li, C., & Cheng, G. (2023). Missdiff: Training diffusion models on tabular data with missing values. arXiv preprint arXiv:2307.00467.](https://arxiv.org/pdf/2307.00467)

---

> > ### Comment · Reviewer_8jb1 · 2024-11-27
> >
> > Thank you for the detailed response! I will maintain my original score.

---

### Official Review · Reviewer_y7yQ · 2024-11-04

**Soundness:** 2
**Presentation:** 3
**Contribution:** 3
**Rating:** 6
**Confidence:** 4

**Summary:**

The paper addresses the imputation of data missing completely at random (MCAR) in tabular data, handling both continuous and categorical variables. The authors propose an EM procedure with a conditional diffusion model for imputation, featuring a novel adaptation of the annealing process for better conditioning on observed values. The paper demonstrates strong results across multiple datasets in comparison with leading methods.

**Strengths:**

* The paper is well-written with a clear, thorough, and concise introduction that effectively summarizes key points from previous works
* The authors specifically address the challenges of the problem and provide clever solution to mitigate them
* The paper's main novelty is supported by theoretical proof
* The evaluations are comprehensive, with thorough and convincing ablation studies

**Weaknesses:**

* A major concern regarding evaluations - while the paper claims to use a single hyperparameter setting throughout, it's unclear how hyperparameters for other methods were selected and their sensitivity to these HP. For me, this concern significantly impacts the overall assessment of the paper.

* While the results are impressive, their importance is not clear. A more convincing evaluation would include the effect on downstream tasks, given imputation is only a first step in most pipelines.

* Another point regarding evaluations, is the sole focus on data missing completely at random. While the MER assumption is important, it is the MNER which is a primary focus in many imputation methods.

* The 0/1 continuous encoding of categorical data is unusual, given that binary data is a known challenge for diffusion models (for example in fields like graph generation). Also, the use of mean is inherently problematic due to common multi-modality in the data

* The novelty of the method compared to other approaches is not clearly articulated in the related works section

### Smaller Issues
* Given the method's novelty isn't specific to tabular data, the related work should include other imputation methods (e.g., image inpainting)
* A simulation study with multiple modes would be valuable, particularly as diffusion-based models should excel in such scenarios
* Despite highlighting the importance of initialization in the EM procedure, the paper doesn't address this point. (Particularly relevant given the naive initial imputation approach)
* It would be interesting to analyze the relationship between delta_t size and ML solution approximation.
* Figure 4 lacks clarity

**Questions:**

In biology, missing values are often represented as 0 (or another "limit of detection" (LOD) value), making it difficult to distinguish between actual LOD values and data missing at random (which can comprise 30% of data in cases like proteomics and single-cell analysis). Do you have any ideas about how this problem could be addressed? Note that the fraction of missing values might be known and could potentially be conditioned on.

---

> ### Author Response · Authors · 2024-11-19
> **Response to Reviewer y7yQ (W1)**
>
> We greatly appreciate the reviewer for taking the time to provide such constructive comments and suggestions. The following are detailed responses to your questions.
>
> ### W1: How the hyperparameters of baseline methods are selected?
> > A major concern regarding evaluations - while the paper claims to use a single hyperparameter setting throughout, it's unclear how hyperparameters for other methods were selected and their sensitivity to these HP. For me, this concern significantly impacts the overall assessment of the paper.
>
> Most of the deep learning baselines recommend the use of one set of hyperparameters for all datasets. For these methods, we directly follow their guidelines and use the default hyperparameters:
>
> * **ReMasker**: we use the recommended hyperparameters provided in Appendix A.2 in  [1]. This set of hyperparameters is searched by tuning on the Letter dataset and is deployed for all the datasets in the original paper; hence, we follow this setting.
> * **HyperImpute**: since HyperImpute works by searching over the space of classifiers/regressors and their hyperparameters, it does not have hyperparameters itself except parameters related to the AutoML search budget. We adopt the default budget parameters of HyperImpute's official implementation for all datasets. The default budget parameters and AutoML search space are provided in https://github.com/vanderschaarlab/hyperimpute/blob/main/src/hyperimpute/plugins/imputers/plugin_hyperimpute.py and Table 5 in the original paper [2].
> *  **MOT and TDM**: There is a main hyperparameter representing the number of subset pairs sampled from the dataset for computing optimal transport loss. Sinkhorn algorithm and TDM are controlled by hyperparameter *n_iter*. While the default value is 3000 (https://github.com/BorisMuzellec/MissingDataOT/blob/master/experiment.py), we set it as 12000 for all datasets to ensure the algorithm converges sufficiently. For the round-robin version of the algorithm, the number of sampled pairs is controlled by *max_iter* and *rr_iter*; we adopt the default value 15, which is enough for the algorithm to converge. For the remaining hyperparameters related to network architectures, we use the default ones for all datasets.
> * **kNN**: we follow the common practice of selecting the number of nearest neighbors as $\sqrt{n}$, where $n$ is the number of samples in the dataset.
> * **GRAPE and IGRM**: we adopt the recommended set of hyperparameters used in the original paper for all datasets. For a detailed explanation of the meaning of the parameters, please see https://github.com/maxiaoba/GRAPE/blob/master/train_mdi.py for GRAPE and https://github.com/G-AILab/IGRM/blob/main/main.py for IGRM.
> * **MissDiff**: since the original implementation is not available, and it is based on diffusion model, for fair comparison, we simply use the same set of hyperparameters with our DiffPuter.
> * **TabCSDI**: We follow the guide for selecting hyperparameters in the original paper (Appendix B in [3]). Specifically, we use a large version of the TabCSDI model with a number of layers set to 4 (see more detailed hyperparameters about the large TabCSDI model at https://github.com/pfnet-research/TabCSDI/blob/main/config/census_onehot_analog.yaml). For batch size, we take the official choice of batch size (8) for the breast dataset (~700 samples) as a base, and scale the batch size accordingly with the sample size of our datasets: since most of the datasets we used have the number of samples between 20000 to 40000, we scale the batch size to 256 and use it for all datasets.
> * **MCFlow**: we adopt the recommended hyperparameters provided in the official implementation for all datasets (https://github.com/trevor-richardson/MCFlow/blob/master/main.py).
>
> For the remaining classical machine learning methods, including EM, GAIN, MICE，Miracle，MissForest, and Softimpute where hyperparameters might be important. Since we use the implementations from the 'hyperimpute' package, we tune the hyperparameters within the hyperparameter space provided in the package (e.g., https://github.com/vanderschaarlab/hyperimpute/blob/main/src/hyperimpute/plugins/imputers/plugin_missforest.py). To be specific, we set the maximum budge as 50, then we sample 50 different hyperparameter combinations according to the hyperparameter space. Finally, we report the optimal performance over the 50 trials.
>
> References:
>
> [1] Tianyu Du, Luca Melis, and Ting Wang. Remasker: Imputing tabular data with masked autoencoding. In International Conference on Learning Representations, 2024.
>
> [2] Daniel Jarrett, Bogdan C Cebere, Tennison Liu, Alicia Curth, and Mihaela van der Schaar. Hyperimpute: Generalized iterative imputation with automatic model selection. In International Conference on Machine Learning, pp. 9916–9937. PMLR, 2022.
>
> [3] Shuhan Zheng and Nontawat Charoenphakdee. Diffusion models for missing value imputation in tabular data. In NeurIPS 2022 First Table Representation Workshop, 2022.

---

> ### Author Response · Authors · 2024-11-19
> **Response to Reviewer y7yQ (W2 & W3)**
>
> ### W2 Importance of missing data imputation. It's effects on downstream tasks.
>
> > While the results are impressive, their importance is not clear. A more convincing evaluation would include the effect on downstream tasks, given imputation is only a first step in most pipelines.
>
> Thank you for your suggestion. We agree with your point that applying well-imputed data to downstream-specific tasks is also very important. Therefore, we considered using the imputed data to perform classification and regression tasks for the target column. We have also added the implementation of this task to our codebase.
>
> Specifically, we first split the complete data into training and test sets, train an XGBoost classifier or regressor on the training set, and test it on the test set. Then, we add masks to the training set to create missing data, and we use different missing value imputation models to obtain imputed data. Next, we train the same XGBoost classifier/regressor using the imputed data and test it on the test set. The effectiveness of imputation can be measured by comparing the performance differences obtained from these two tests.
>
> In the table below, we show the performance on the test set when training on complete data, as well as training on imputed data obtained through different imputation methods.
>
>
> | Datasets | Metric | Real | -  | DiffPuter | HyperImpute | ReMasker | MOT
> | -------  | ------ | ------| ---|  ------ | ------ | ----- | ---- |
> | Adult | AUC	| 0.9270 | $\uparrow$ | **0.9252** | 0.9235 | 0.9218 | 0.9219 |
> | Shoppers | AUC | 0.9300 | $\uparrow$ | **0.9255** | 0.9132 | 0.9067 | 0.9251 |
> | Beijing | RMSE | 0.1205 | $\downarrow$ | **0.1504** | 0.1583 | 0.1543 | 0.1537 |
>
> As demonstrated in the table, our method still outperforms other SOTA imputation methods on this task. However, we find that the performance differences between different methods are not very large, therefore this task might not be sufficient to evaluate the quality of imputation comprehensively. We will appreciate it if you can suggest more appropriate downstream tasks.
>
> ### W3: Evaluation on other missing settings (e.g., MNAR).
>
> > Another point regarding evaluations, is the sole focus on data missing completely at random. While the MER assumption is important, it is the MNER which is a primary focus in many imputation methods.
>
> Regarding MER and MNER mentioned by the Reviewer, we assume that these should be MAR (Missing At Random) and MNAR (Missing Not At Random). We humbly point out that our experimental section considered all three missing mechanisms: 1) Missing Completely At Random (MCAR), 2) Missing At Random (MAR), and 3) Missing Not At Random (MNAR). This is explained in Section 5.1, Datasets part. Due to length constraints in the main text, we only presented the experimental results for the MCAR setting. We placed the specific implementation of these three settings in Appendix C.3 (D.3 in the revised version of paper), and the experimental results in Appendix D.2 and Appendix D.3. (E.2 and E.3 in the revised paper).

---

> ### Author Response · Authors · 2024-11-19
> **Response to Reviewer y7yQ (W4 & W5)**
>
> ### W4: Encoding for categorical data.
> > The 0/1 continuous encoding of categorical data is unusual, given that binary data is a known challenge for diffusion models (for example, in fields like graph generation).
>
> One-hot encoding is a conventional method for handling discrete/categorical tabular data, and has been used in many previous papers [1,2,3], although it may not be the most appropriate one. We acknowledge that how to more effectively handle categorical data is indeed a major challenge in tabular deep learning, but it is out-of-scope of this paper. Essentially, one-hot encoding is just the simplest choice for processing categorical data, and if there are other more reasonable methods, they can be directly combined with our approach.
>
> References:
>
> [[1] Kim, Jayoung, Chaejeong Lee, and Noseong Park. "STaSy: Score-based Tabular data Synthesis." The Eleventh International Conference on Learning Representations.](https://arxiv.org/abs/2210.04018)
>
> [[2] Zheng, Shuhan, and Nontawat Charoenphakdee. "Diffusion models for missing value imputation in tabular data." arXiv preprint arXiv:2210.17128 (2022).](https://arxiv.org/pdf/2210.17128)
>
> [[3] Liu, Tennison, et al. "GOGGLE: Generative modelling for tabular data by learning relational structure." The Eleventh International Conference on Learning Representations. 2023.](https://openreview.net/forum?id=fPVRcJqspu)
>
> > Also, the use of mean is inherently problematic due to common multi-modality in the data.
>
> For numerical data, mean imputation is natural. For categorical data, our mean imputation based on one-hot encoding essentially estimates the missing column values according to the marginal distribution (prior distribution) of each category. For example, for the gender column, if the data contains 80% male and 20% female, then for a data point with a missing gender column, we naturally assume it has an 80% probability of being male and a 20% probability of being female, therefore assigning its corresponding (one-hot) embedding vector as [0.8, 0.2]. This is a natural and reasonable processing approach.
>
> Furthermore, from the experimental results, our simple processing method has already achieved quite good results (outperforming other baseline methods).
>
> ### W5: Clarify the novelty of the proposed method in the related works section.
> > The novelty of the method compared to other approaches is not clearly articulated in the related works section.
>
> We thank the Reviewer for their suggestion. In the last paragraph of Related Works, we implicitly pointed out the novelty of our method by summarizing the limitations of existing methods. In the revised paper, we have added the following sentence, which clearly and explicitly states the novelty of our method compared with existing ones:
>
> "By contrast, the proposed DiffPuter is the first to integrate a diffusion-based generative model into the EM framework. Additionally, we achieved accurate conditional sampling by mixing the forward and reverse processes of diffusion and demonstrated the effectiveness of this approach through theoretical analysis."

---

> ### Author Response · Authors · 2024-11-19
> **Response to Reviewer y7yQ (Q1)**
>
> ### Q1: How to address "missing data as limit of detection" (LOD) problem?
>
> > In biology, missing values are often represented as 0 (or another "limit of detection" (LOD) value), making it difficult to distinguish between actual LOD values and data missing at random (which can comprise 30% of data in cases like proteomics and single-cell analysis). Do you have any ideas about how this problem could be addressed? Note that the fraction of missing values might be known and could potentially be conditioned on.
>
> Thank you for raising this interesting and worthy question for discussion. We are happy to share some of our thoughts on addressing this issue.
>
> First, it is obvious that if we have only one column in our data, where both LOD data and missing data are forcibly observed as 0, this would indeed create a fundamental identification problem, specifically manifested in:
>
> Assuming the real data follows a certain distribution (such as normal distribution)
> When we observe a value of 0, we cannot distinguish whether this 0 comes from:
>
> - Real value ≤ 0 being truncated
> - Completely random missing.
>
> At the point of 0, the observed results produced by these two mechanisms are exactly the same. However, since our data is multivariate (having many different columns), and the values in other columns may influence whether the target column's value is LOD or missing. Therefore, we can obtain additional information from these to help us identify whether a column is LOD or missing.
>
>
> For example, consider the following simple scenario. Besides the target column (denoted as $x_{j_1}$), we have another column $x_{j_2}$ that is fully observed, and $x_{j_1}$'s observed values have a strong correlation with $x_{j_2}$. For instance, when $y \ge 0$, $x_{j_1}$'s values tend to concentrate near LOD = 0, while when $y$ < 0, $x_{j_1}$'s values are typically much larger than 0. Then, if we obtain an $x_{j_1} = 0$ and $x_{j_2}$ > 1, we can reasonably conclude that the $x_{j_1}$'s value is indeed the observed LOD value, rather than a missing value.
>
> Extending to more general cases, if the value of the target column $x$ (whether it is missing or not) is indeed conditioned on other columns, we can train a model to identify whether it is an observed LOD value or missing. We can assume that this model can output a probability $p$, representing a $p$ probability that the position is a missing value, and a $1-p$ probability that it is an observed LOD value.
>
>
> Consider a tabular dataset $X \in \mathbb{R}^{N \times d}$, and $x_{ij}$ denotes the value at row $i$ and column $j$. Let's assume $LOD = 0$ for convenience. Then, we may resort to statistic models, e.g., mixture models, to handle this problem.
>
> For example, we may consider the mixture of Gamma and Gaussian distributions. For each non-zero observation $x_{ij}$ in position $(i,j)$:
> $P(x_{ij}) = w_j P_{\text{gamma}}(x_{ij}|\alpha_j,\beta_j) + (1-w_j) P_{\text{gaussian}}(x_{ij}|\mu_j,\sigma_j^2)$
>
> where:
> $P_{\text{gamma}}(x|\alpha,\beta) = \frac{\beta^\alpha}{\Gamma(\alpha)} x^{\alpha-1} e^{-\beta x}$,
> $P_{\text{gaussian}}(x|\mu,\sigma^2) = \frac{1}{\sqrt{2\pi\sigma^2}} e^{-\frac{(x-\mu)^2}{2\sigma^2}}$
>
>
> The log-likelihood for column $j$:
>
> $\mathcal{L}_j = \sum\_{i, x\_{ij} \neq 0} \log [w_j P _{gamma}(x _{ij}|\alpha _j,\beta _j)]$
>
> $ + (1-w_j) P_{\text{gaussian}}(x_{ij}|\mu_j,\sigma_j^2)] + \sum\_{i:x_{ij}=0} \log[w_j P_{\text{gamma}}(\varepsilon|\alpha_j,\beta_j) + (1-w_j) P_{\text{gaussian}}(0|\mu_j,\sigma_j^2)]$
>
> The optimal distribution parameters can be obtained via Maximum likelihood estimation:
> $\hat{\alpha} _j,\hat{\beta} _j, \hat{\mu} _j, \hat{\sigma} _j^2, \hat{w} _j = \arg\max _{\alpha _j,\beta _j,\mu _j,\sigma _j^2,w _j} \mathcal{L}_j$
> subject to the following conditions:
>
> - $\alpha_j, \beta_j, \sigma_j > 0$
> - $0 \leq w_j \leq 1$
>
> The above approach assumes that the missing probability only relates to the column and not to the row. To consider the influence of relationships between rows, we can first cluster the input data, and for each cluster, learn a cluster-specific mixture model.
>
>
> Finally, for zero values, the missing probability is:
> $P(\text{miss}|x_{ij}=0) = \frac{(1-w_j)P_{\text{gaussian}}(0|\mu_j,\sigma_j^2)}{w_j P_{\text{gamma}}(\varepsilon|\alpha_j,\beta_j) + (1-w_j)P_{\text{gaussian}}(0|\mu_j,\sigma_j^2)}$
>
> Our method can be effective based on this foundation. For each position marked as an LOD value ($=0$), there is a probability $p$ where we assume it is missing, and using our method we ultimately obtain a predicted imputation value, which we denote as $x_{pred}$. Additionally, there is a $1-p$ probability where we consider it to be the observed LOD value. Therefore, we mix these two scenarios according to their probabilities, meaning the final predicted value is $p * x_{pred} + (1-p) * 0$.

---

> ### Author Response · Authors · 2024-11-20
> **Response to Reviewer y7yQ (other issues)**
>
> ### Other issues
>
> > Given the method's novelty isn't specific to tabular data, the related work should include other imputation methods (e.g., image inpainting)
>
> Thank you for the suggestion! We've included several related works about image inpainting in Section 2 of the revised paper.
>
>
> > A simulation study with multiple modes would be valuable, particularly as diffusion-based models should excel in such scenarios.
>
> We don't quite get what you mean by "simulation study of multiple modes". We would be very grateful if you could provide clearer hints about this.
>
>
> > Despite highlighting the importance of initialization in the EM procedure, the paper doesn't address this point. (Particularly relevant given the naive initial imputation approach)
>
> How to initialize missing values is indeed important. In the paper, we only adopted the simplest approach of using the mean of observed values for initialization and have already achieved good results. Considering the convergence property of the EM algorithm, developing novel initialization methods might not bring significant improvements in performance but might improve training speed (via improving the convergence speed).
>
> We've also tried to initialize the missing values using KNN (which is a baseline in the experiments). The table below compares the training time and MAE score obtained without and with KNN as the base imputation method, respectively, on the adult dataset.
>
> | Adult dataset | KNN | Diffputer(vanilla), 5 iterations | DiffPuter (init with KNN), 3 iterations |
> | ------ | --------- | ------ | --------- |
> | Training Time | 41.5s  | 2142.9s | 1358.6s |
> | MAE           | 0.5520 | 0.4853  | 0.4839  |
>
> As demonstrated in the Table, using a KNN model as the initial imputer can greatly reduce the training time with marginal performance improvement.

---

> > ### Comment · Reviewer_y7yQ · 2024-11-23
> > **Rebuttal Response**
> >
> > Thank you for the detailed response and sorry for the late reply!
> >
> > Thank you very much for the clarifications about the hyperparameter tuning of the compared methods, this is most useful.
> >
> > About the downstream tasks, I believe more "classical" downstream tasks which are known to be highly affected by imputation, like with genomic data, might be too involved for this step.
> > Also, thank you for the detailed answer regarding the LOD case. I agree that your suggestion makes sense and I hope you will be able to further investigate it on real data in the future.
> >
> > With that, I still believe the comparison to other methods is lacking and it's not clear what distinguishes your method from other methods or other generative modeling approaches (as other reviewers suggested).
> >
> > Following all of the above, given the impressive empirical results, I will raise my score.

---

> > > ### Author Response · Authors · 2024-11-24
> > > **Thank you very much for your feedback!**
> > >
> > > We thank the reviewer for acknowledging the contributions of our work. Regarding the issue you raised about insufficient empirical comparison with other methods, we will submit comparisons with more deep generative models in the new version (as suggested by Reviewer TxpG). Thank you again for your insightful comments and constructive suggestions.

---

### Author Response · Authors · 2024-11-23
**General Response to All Reviewers**

We thank all reviewers for their professional review and valuable opinions and suggestions. Regarding the questions you raised, we have provided detailed answers one by one in our response for your review.

Furthermore, we have uploaded the revised paper, and here, we summarize the updated content.

**Reviewer y7yQ**

- In Section 2's Related Works, we have added an introduction to imputation methods from other fields, and explained the differences between our proposed DiffPuter and existing methods.
- In Section 5, we moved the training time comparison experiments from the Appendix to Section 5.3, considering their importance.
- In Appendix D.6, we have detailed how the hyperparameters of the baseline methods were selected and set.

**Reviewer 8jb1**

- In Section 5, we moved the training time comparison experiments from the Appendix to Section 5.3, considering their importance.

**Reviewer TxpG**
- In Section 1, we have provided a clearer explanation of this paper's motivation. We specifically elaborated on the disadvantages of other Deep Generative Models in performing tabular data density estimation (i.e., M-step), and the irreplaceable advantages of Diffusion models in learning tabular data distribution.
- In Section 5, we have provided more literature support for the strong imputation performance of traditional machine learning methods. In addition, we moved the training time comparison experiments from the Appendix to Section 5.3, considering their importance.

**Reviewer rwwt**
- In Section 4, we have refined and supplemented some specific details of our method (such as $\sigma(t)$) and theoretical analysis.
- In Section 5, we moved the training time comparison experiments from the Appendix to Section 5.3, considering their importance.
- We have added a new Appendix A, which introduces the Symbols used in this paper.
- In Appendix D.3, we have added the specific implementation details for the two missing mechanisms, MAR and MNAR.
- In Appendix D.6, we have detailed how the hyperparameters of the baseline methods were selected and set.

We hope that the above revisions can further address your concerns. Thank you again for your diligent effort in reviewing.

---

### Meta-Review · Area_Chair_mSHH · 2024-12-16

**Metareview:**

The paper addresses the problem of missing-data imputation in tabular data with a conditional diffusion model algorithm (DiffPuter) that corresponds to an expectation-maximization approach. A strong empirical evaluation is provided. All the reviews recommend acceptance and I agree.

**Additional Comments On Reviewer Discussion:**

All the reviews recommend acceptance and I agree.

---

### Decision · Program_Chairs · 2025-01-22

Accept (Spotlight)